# Ultrahigh thermoelectricity obtained in classical BiSbTe alloy processed under super-gravity

Min Zhou [1,9] ✉, Haojian Su[1,2,9], Jun Pei [3], Li Wang[4], Hualu Zhuang [5], Jing-Feng Li [5] ✉, Kun Song [6], Haoyang Hu[6], Jun Jiang [6] ✉, Qinghua Zhang [7], Jiangtao Li [8] & Laifeng Li [1] ✉

Thermoelectric materials allow direct conversion between heat and electricity and may be useful for power generation or solid-state refrigeration. However, improving thermoelectric performance is challenging because of the strong coupling between the electrical and thermal transport properties. We demonstrate a new super-gravity-field re-melting fabrication technology that synergistically optimizes the thermoelectric performance. Using a super-gravity field, the brittle $(Bi,Sb)_2Te_3$ alloy undergoes unusual plastic deformation and forms mounts of microstructure defects, which is rarely observed in common fabrication process. As a result, the microstructure reconstruction and carrier concentration optimization were simultaneously realized, resulting in an ultra-low lattice thermal conductivity of <0.25 W/m K and a record-high figure of merit of >1.91 in the BiSbTe alloy. The strong enhancement of thermoelectric properties was validated in a thermoelectric module with high conversion efficiency of 6.4% and corresponding output power density of 0.34 W/cm² when subjected to a temperature difference of 185 K. This work highlights a new super-gravity strategy to achieve a high thermoelectric performance, which may be applicable to other thermoelectric materials.

Thermoelectric materials interconvert electrical and thermal energy and have potential applications for waste heat power generation and all-solid-state refrigeration based on the Seebeck effect and Peltier effect, respectively[1,2]. The widespread use of thermoelectric materials is limited by their low thermoelectric conversion efficiencies. These are characterized by the dimensionless figure of merit $zT = \alpha^2\sigma T/(\kappa_L + \kappa_e)$, where $\alpha$, $\sigma$, $\kappa_L$, $\kappa_e$, and $T$ are the Seebeck coefficient, electrical conductivity, lattice thermal conductivity, electronic contribution to thermal conductivity, and absolute temperature, respectively. $\alpha^2\sigma$ is usually called the power factor (PF). It is difficult to simultaneously manipulate electrical and thermal transport performance due to the complex coupling between these parameters. In recent years, studies have focused on optimizing the thermoelectric properties using band engineering[3,4], microstructure engineering[5,6], and by implementing new fabrication technologies[7-9]. These approaches have resulted in higher $zT$ values in some thermoelectric materials.

[1]State Key Laboratory of Cryogenic Science and Technology, Technical Institute of Physics and Chemistry, Chinese Academy of Sciences, Beijing, China. [2]Centre of Materials Science and Optoelectronics Engineering, University of Chinese Academy of Sciences, Beijing, China. [3]Beijing Municipal Key Laboratory of New Energy Materials and Technologies, School of Materials Science and Engineering, University of Science and Technology Beijing, Beijing, China. [4]School of Mechanical Engineering, Tianjin Sino-German University of Applied Science, Tianjin, China. [5]State Key Laboratory of New Ceramic Materials, School of Materials Science and Engineering, Tsinghua University, Beijing, China. [6]Ningbo Institute of Materials Technology and Engineering, Chinese Academy of Sciences, Ningbo, China. [7]Institute of Physics, Chinese Academy of Sciences, Beijing, China. [8]Suzhou Institute for Advanced Research, University of Science and Technology of China, Suzhou, China. [9]These authors contributed equally: Min Zhou, Haojian Su. ✉e-mail: mzhou@mail.ipc.ac.cn; jingfeng@mail.tsinghua.edu.cn; jjun@nimte.ac.cn; lfli@mail.ipc.ac.cn

$Bi_2Te_3$-based alloys are the most widely used commercial thermoelectric materials for applications at room temperature[10]. Although a lot of new thermoelectric materials have been discovered and received more attention, $Bi_2Te_3$-based alloys remain at the forefront of thermoelectric research. $Bi_2Te_3$ is a remarkably good thermoelectric material, naturally having low lattice thermal conductivity and complex electronic structure. Commercial $Bi_2Te_3$-based materials are typically prepared by zone melting with peak $zT$ value of around unity for tens of years. This limitation restricts their applications to niche areas because of their low efficiency compared with those of other energy conversion materials, highlighting the need to improve their thermoelectric properties. Innumerable research efforts have focused on improving the thermoelectric performance of $Bi_2Te_3$-based alloys. The band structure engineering is an effective approach in the enhancement of thermoelectric performance for $Bi_2Te_3$. Solid solution alloying with $Sb_2Te_3$ or $Bi_2Se_3$ induces band convergence, which increases the density of states as well as reduces lattice thermal conductivity[11]. In the p-type $(Bi,Sb)_2Te_3$ system, valence band convergence occurs near the most commonly used composition $Bi_{0.5}Sb_{1.5}Te_3$ and the corresponding peak $zT$ value is about 1 (300 K)[12]. Sn impurity in the valence band of $Bi_2Te_3$ enhances Seebeck coefficient through resonant scattering[13,14]. Au doping on the Bi site of $Bi_2Te_{2.7}Se_{0.3}$ also induces resonant states, leading to increase of the Seebeck coefficient. Peak $zT$ value of 0.91 is obtained for $Cu_{0.008}Bi_{1.99}Au_{0.01}Te_{2.7}Se_{0.3}$ (320 K)[15]. CuI-doped $(CuI)_xBi_2Te_{2.7}Se_{0.3}$ increases the crystalline electric field, which results in the Rashba band splitting. The formation of Rashba band effect enhances the PF and $zT$ value in a wide temperature range[16,17]. The point defect engineering is also effective to optimize thermoelectric properties of $Bi_2Te_3$-based alloys. Antisite defects ($Bi'_{Te}$, $Sb'_{Te}$) and donor-like effects are engineered by tuning the formation energy of point defects[18]. Recent studies have focused on investigating structural modification to enhance the $zT$ values of polycrystalline $Bi_2Te_3$ alloys using different processing methods[19–21], such as ball-milling and hot-pressing[19,22,23], melt-spinning and spark-plasma-sintering[8,24], hot-forging[25], low-temperature hydrothermal and hot-pressing treatment[26]. A higher $zT$ of 1.4 (373 K) was reported in nanocrystalline BiSbTe alloy[19]. However, no significant increase in $zT$ values was realized until *Science* published another work, which obtained a high $zT$ of 1.86 in $(Bi,Sb)_2Te_3$ compounds via liquid phase sintering with excess $Te$[27]. It is a pity that this work was controversial and could not be repeated in nearly ten years. Jo[28] and Deng[29,30] even constructed the similar microstructure in the Te-rich $Bi_{0.5}Sb_{1.5}Te_3$ alloys, the obtained maximum $zT$ value was just 1.2–1.3, which was much lower than that reported by Kim[27]. So, high-performance $Bi_2Te_3$-based thermoelectric materials are expected.

## Results and discussion

In the present work, we develop a new fabrication technology, super-gravity-field re-melting (SGF-RM) (Fig. S1), to realize high-performance $(Bi,Sb)_2Te_3$ thermoelectric material with record-high figure of merit of >1.91. Using a super-gravity field, the $(Bi,Sb)_2Te_3$ raw material was melted in a "chemical furnace" and then quickly solidified. The equivalent super-gravity field ($G$) is induced by the high-speed rotation of rotors and is expressed as $G = \omega^2 L$, where $\omega$ is the angular velocity and $L$ is the distance from the axis of rotation to the point of interest. A self-propagating "chemical furnace" (strong exothermic reaction, Ti $+2B \rightarrow TiB_2$, 66.8 kcal mol$^{-1}$)[30], which replaces the traditional high-temperature melting furnace, is used to melt the raw materials. When the super-gravity reaches a set value, the chemical furnace is ignited. During the burning process, a large amount of heat energy is created, which melts the raw materials. After the combustion reaction is complete, the melts quickly cool and solidify under super-gravity (Figs. 1a and S2).

Compared with traditional melting, the mass and heat transfer of the melts in the super-gravity field were faster than those occurring in the Earth-gravity field. The removal velocity of bubbles in the melt was strongly correlated with the supergravity coefficient ($G/g$, g = 9.8 m/s$^2$) and the bubble radius[31,32] (Figs. 1b and S3). Small bubbles were difficult to remove from the melt during the short solidification process due to their lower removal velocities (Table S1), which promoted the formation of micropores in the obtained bulks. Under super-gravity, the material underwent a rapid volume change and plastic deformation during solidification. This effect is rarely observed in most brittle thermoelectric materials for conventional deformation processes. The plastic deformation process readily induced high-density dislocations in the alloys, and the super-gravity increased the degree of super-cooling, accelerated the solidification rate, and refined the crystal grains. Thus, the distinctive SGF-RM technology facilitates the reconstruction of microstructures via changing the solidification process of $(Bi,Sb)_2Te_3$ melts. Furthermore, the SGF-RM is efficient and economical, showing great potential for future industrial applications.

In fact, the study on fabrication of metals[33] and ceramics[34,35] under high gravity field has been reported in the past decades. Until recently, highly-dense $Cu_2ZnSnSe_4$[36], SnTe-based[37,38] thermoelectric materials we successfully synthesized under high gravity field. Notably, SGF-RM facilitated microstructure reconstruction by changing the solidification process of the melt (Fig. 1c). The higher number of grain boundaries and micropore interfaces scattered mid/long-wavelength phonons, which reduced the lattice thermal conductivity. The absence of a conduction medium in the micropores also decreased the thermal conductivity, although it reduced the carrier mobility[6,39]. The introduced high-density dislocations targeted short and medium-wavelength phonons, which reduced the lattice thermal conductivity. Furthermore, the enhanced point defects after SGF-RM (discussed later) targeted short-wavelength phonons. Thus, a full-spectrum strategy targeting a wide spectrum of phonons was realized, resulting in an ultra-low lattice thermal conductivity of <0.25 W/m K at 300 K (Fig. 1e). However, excess Te tended to evaporate from the melt due to its lower vapor pressure, which generated anti-site defects because Bi(Sb) occupied Te vacancies during the melting process under super-gravity. This increased the carrier concentration and PF (Fig. 1d and f). Benefitting from the improved PF and a significant decrease in the lattice thermal conductivity, high $zT$ value of over 1.91 (375 K) was obtained for the re-melted $(Bi,Sb)_2Te_3$ alloy under super-gravity (Fig. 1g), which was higher than many reported data[6,7,19,27,40–42]; this finding was confirmed and reproducible (Figs. S7 and S8). These results suggest that SGF-RM is an efficient method for processing high-performance thermoelectric materials.

The $(Bi,Sb)_2Te_3$ alloy with a rhombohedral structure showed anisotropic thermoelectric properties (Figs. 2, S2, and S9), in which greater thermoelectric properties were obtained in the cross-plane direction (Figs. 2 and S2). The electrical conductivity decreased monotonically, indicating a degeneration in the material's semiconductor characteristic. The Seebeck coefficient exhibited an initial increase and then a decrease with the temperature, which was associated with intrinsic excitation[43,44]. After SGF-RM, the electrical conductivity increased, but the Seebeck coefficient decreased, which was related to the increased carrier concentration (Table 1). Besides, the increased carrier concentration inhibited the intrinsic excitation[45,46] and pushed the peak Seebeck coefficient to higher temperatures (Fig. 2b). So, the trend of Seebeck coefficient changed at higher temperatures after SGF-RM. Due to the enhanced electrical conductivity and slightly lower Seebeck coefficient, the PF increased over the measured temperature range of 300–500 K and reached a maximum of 44.5–48.9 μW/K$^2$ cm at 300 K; this was about 11%–22% higher than that of the $(Bi,Sb)_2Te_3$ alloy before SGF-RM (Fig. 2c). The total thermal conductivity ($\kappa$) and lattice thermal conductivity ($\kappa_L$) showed similar temperature dependences (Fig. 2d, e), indicating that the lattice thermal conductivity made a marked contribution to the total thermal conductivity. As the temperature increased, the $\kappa_L$ initially decreased,

due to Umklapp scattering, and then increased as intrinsic excitations occurred and dominated the transport process. The thermal conductivity and lattice thermal conductivity decreased after SGF-RM. An ultra-low lattice thermal conductivity of 0.15–0.25 W/m K was obtained, which approached the amorphous limit calculated by the Cahill model[47,48] (Fig. 2e). Benefiting from the increased PF and reduced thermal conductivity, the peak $zT$ of 1.91–1.97 (Fig. 1g) and average $zT$ of 1.63–1.66 (Fig. 2f) were obtained for the $Bi_{0.48}Sb_{1.52}Te_{3.03}$ alloy after SGF-RM.

To further investigate the electrical transport properties of the $(Bi,Sb)_2Te_3$ alloys, the carrier concentration and mobility were discussed in detail. After SGF-RM, the carrier concentration increased, but the mobility decreased (Table 1). An enhanced carrier concentration has also been reported for $(Bi,Sb)_2Te_3$-based alloys synthesized by melt-spinning compared with those prepared by the traditional melting/quenching/annealing synthesis[40]. The $(Bi,Sb)_2Te_3$ alloys were $p$-type semiconductors whose carrier concentrations were affected by anti-site defects and anion vacancies. During the melting process under super-gravity, excess telluride atoms tended

to evaporate from the melt due to their lower vapor pressure, which was confirmed by inductively coupled plasma–optical emission spectroscopy (ICP-OES) (Table 1). Owing to their similar atom radii and physical properties, Bi(Sb) atoms mainly occupied the Te sites to form $Bi'_{Te}$ or $Sb'_{Te}$ anti-site defects (Fig. 1d)[49], where $V^{\bullet\bullet}_{Te}$ is a Te vacancy, $V'''_{Bi}$ or $V'''_{Sb}$ is a Bi or Sb vacancy, $Bi'_{Te}$ or $Sb'_{Te}$ is an anti-site defect, and $h^{\bullet}$ is a hole. Thus, the evaporation of excess Te in the $Bi_{0.48}Sb_{1.52}Te_{3.03}$ alloy generated more anti-site defects when Bi(Sb) atoms occupied Te vacancies[50]. These negatively charged anti-site defects formed extra holes in the matrix, which increased the carrier concentration (Table 1). As a result, the electrical conductivity increased, but the Seebeck coefficient decreased after SGF-RM. The carrier mobility decreased after SGF-RM, primarily due to increased carrier–carrier scattering and enhanced microstructural defects scattering, which would be discussed in the following. A little Te volatilization after SGF-RM resulted in increased carrier concentration but did not obviously change the band structure of the BiSbTe compound. The measured band gap showed consistency before and after SGF-RM (Fig. S10).

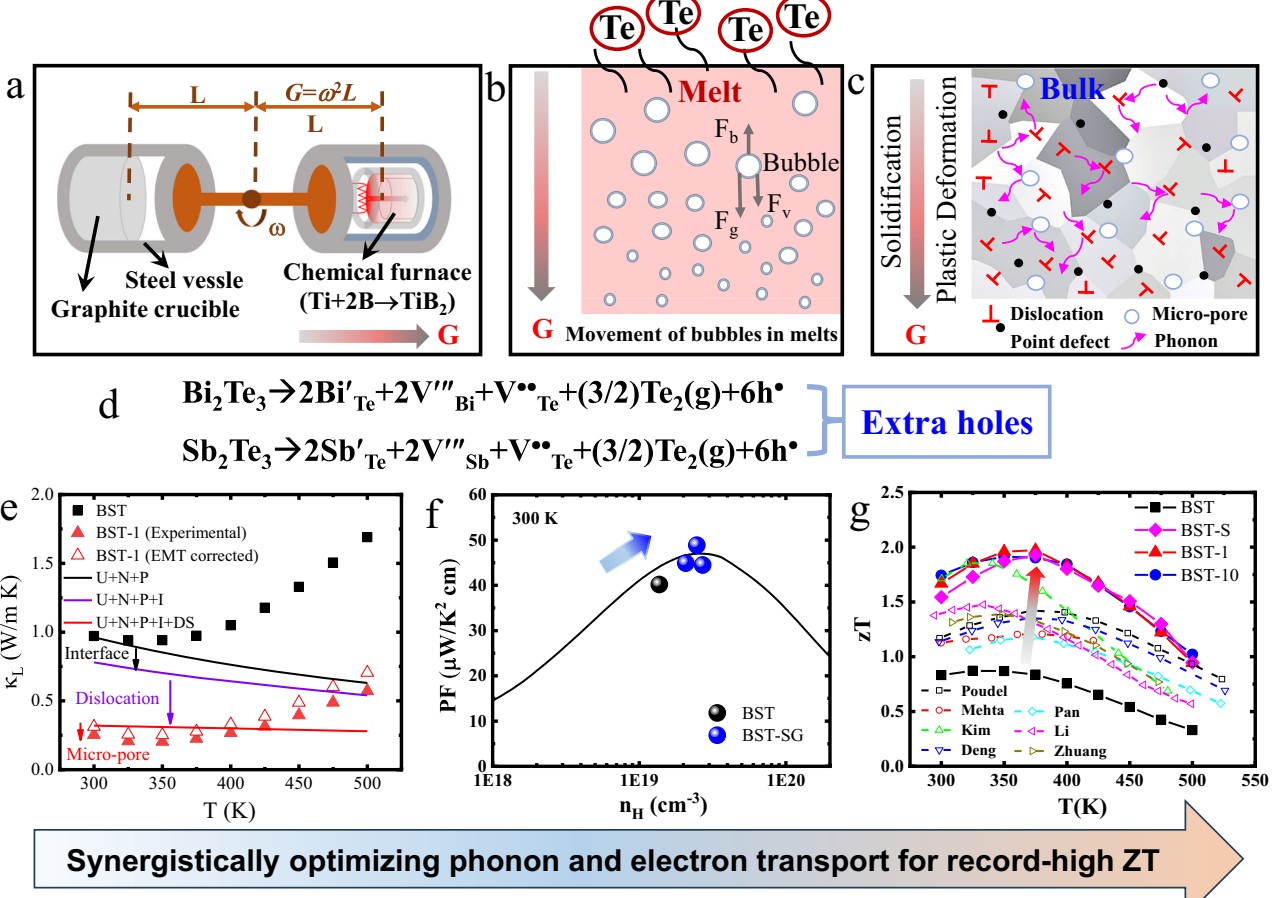

**Fig. 1 | Synergistically optimizing phonon and electron transport for record-high $zT$ values.** Schematic illustration of the **a** super-gravity-field re-melting technology, **b** movement of bubbles in melts, and **c** reconstruction of microstructures after super-gravity-field re-melting (SGF-RM). **d** Process of Te evaporation causing extra holes. **e** Lattice thermal conductivities ($\kappa_L$) of samples before and after SGF-RM. The solid symbols present the experimental results. The black solid line represents the predicted $\kappa_L$ value considering the scattering of the Umklapp process, normal process, and point defects (U + N + P). The purple solid line represents the predicted $\kappa_L$ value considering the additional scattering of grain boundaries and micro-pore interfaces (U + N + P + I). The red solid line represents the predicted $\kappa_L$ values considering the additional scattering of dislocations (U +

N + P + I + DS). The effective medium theory (EMT)-corrected values are shown by red empty triangles. **f** Power factor values as a function of the Hall carrier concentration predicted by the effective mass $m^* = 1.05\, m_0$ and drift mobility $\mu_w = 420\, cm^2/V\,s$ at 300 K. **g** $zT$ values of the $Bi_{0.48}Sb_{1.52}Te_{3.03}$ alloy before (BST) and after SGF-RM. Note: The sample with hand-milled powders is denoted as BST-1 after re-melting under super-gravity for 1 min. The sample with hand-milled powders is denoted as BST-10 after re-melting under super-gravity for 10 min. The sample with particle sizes between 0.6 and 1 μm is denoted as BST-S after re-melting under super-gravity for 10 min. Some reported typical results of $(Bi,Sb)_2Te_3$-based materials are also shown in this figure (**g**) [6,7,19,27,40–42].

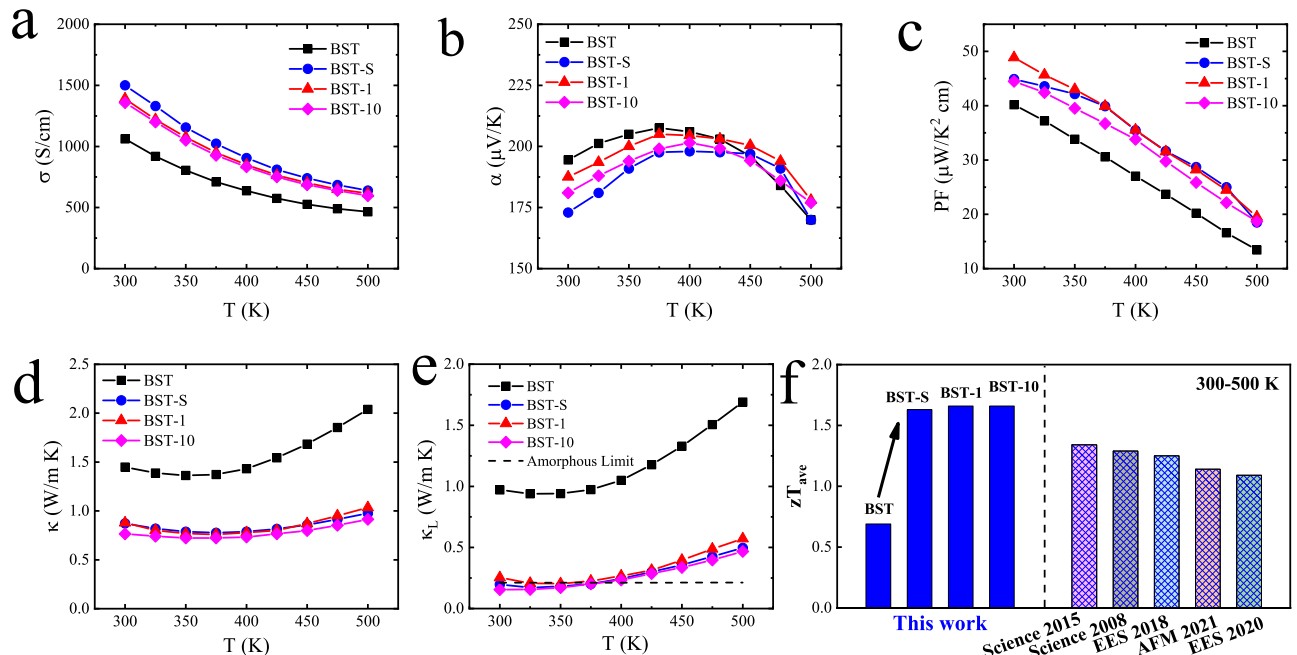

**Fig. 2 | Thermoelectric properties.** Temperature dependence of **a** electrical conductivity, **b** Seebeck coefficient, **c** power factor, **d** thermal conductivity, **e** lattice thermal conductivity, and **f** average $zT$ values of the $(Bi,Sb)_2Te_3$ alloys before and after SGF-RM. Some data of previously reported typical $(Bi,Sb)_2Te_3$-based materials are also shown in this figure (**f**) [7,19,27,40,41].

## Table 1 | The ICP-OES results, the corresponding Hall measurement of all the samples before and after SGF-RM

| samples | ICP-OES results | Carrier concentration ($10^{19}$ cm$^{-3}$) | Mobility (cm$^2$/V S) | Hall coefficient (cm$^3$/C) | Electrical resistivity ($10^{-3}$ Ω cm) |
|---------|-----------------|---------------------------------------------|-----------------------|-----------------------------|------------------------------------------|
| BST | $Bi_{8.51}Sb_{28.92}Te_{62.57}$ | 1.37 | 388.98 | 0.46 | 1.17 |
| BST-S | $Bi_{9.99}Sb_{29.27}Te_{60.74}$ | 2.08 | 279.01 | 0.30 | 1.03 |
| BST-1 | $Bi_{10.20}Sb_{28.45}Te_{61.35}$ | 2.42 | 358.62 | 0.26 | 0.72 |
| BST-10 | $Bi_{8.57}Sb_{29.12}Te_{62.30}$ | 2.69 | 315.06 | 0.23 | 0.74 |

## Table 2 | Positron annihilation lifetime spectroscopy (PALS) data of BST, BST-1, and BST-1-R

| Specimen | | $\tau_1$ (ns) | $I_1$ (%) | $\tau_2$ (ns) | $I_2$ (%) | $\tau_3$ (ns) | $I_3$ (%) |
|----------|--------|---------------|-----------|---------------|-----------|---------------|-----------|
| BST | | 0.1521 | 26.3 | 0.3209 | 71.6 | 1.286 | 2.06 |
| BST-1 | | 0.1685 | 31.1 | 0.3344 | 67.7 | 1.534 | 1.25 |
| BST-1-R | Top | 0.2076 | 60.90 | 0.3922 | 38.4 | 1.870 | 0.70 |
| BST-1-R | Middle | 0.2076 | 60.91 | 0.3870 | 38.4 | 1.762 | 0.69 |
| BST-1-R | Bottom | 0.2056 | 57.98 | 0.3758 | 41.3 | 1.765 | 0.73 |

In this work, an effective mass model was also used to evaluate the charge carrier transport properties (**Effective mass modeling in Supporting information**)[6,51]. Although bipolar effects easily occur in $Bi_2Te_3$-based alloys, a single-parabolic-band model was used in this work. This model assumed that minor charge carriers contributed little to the electrical conductivity[6,7]. According to the fitted curves with the assumed effective mass $m^* = 1.05\, m_0$ ($m_0$ is the inertial mass of a free electron) and the drift mobility $\mu_0 = 420$ cm$^2$/V s, the predicted Hall-carrier-concentration-dependent Seebeck coefficient and Hall mobility were obtained (Fig. S11). The results confirmed the validity of the single-parabolic-band model.

Findings from recent studies[6,7,27,40,52,53] are consistent with those found in this work. The results showed some discrepancies from the fitting lines for the $(Bi,Sb)_2Te_3$ materials, which may have been related to the complex electronic structures of the $(Bi,Sb)_2Te_3$ alloys. However, the weighted mobility $\mu_w$, which characterizes the drift mobility

and inherent transport properties, was consistent between the samples before and after SGF-RM (Fig. S11c), further confirming that the acoustic phonon scattering mechanism was unchanged. With a combination of the predicted Hall-carrier-concentration-dependent Seebeck and Hall mobility, the thermoelectric PF was calculated and compared with the experimental data (Fig. S11d). The comparison suggested that the improvement in the PF was mainly due to the optimization of the carrier concentration after SGF-RM.

Positrons are sensitive and self-seeking probes for microstructural defects, and positron annihilation measurements provide a way to qualitatively analyze anti-site defects, dislocations, vacancies, and even pores inside materials[54,55]. The measured positron annihilation spectra (Fig. S12) were decomposed into three lifetimes, $\tau_1$, $\tau_2$, and $\tau_3$, with corresponding intensities $I_1$, $I_2$, and $I_3$, respectively, using the LT9.0 software (Table 2). The longest-lifetime component $\tau_3$ may have been due to the annihilation of the ortho-positronium formed on the surfaces of the specimens and/or some low-energy positrons annihilated by inner $^{22}$Na[56]. As the values of the relative intensity $I_3$ of the samples was the weakest (< 2.5%), it will not be discussed in this paper. The positron lifetime $\tau_1$ represents the free positron lifetime originating from anti-site defects, dislocations, and small vacancies, while $\tau_2$ was likely caused by large clusters of vacancies and micropores[57]. Notably, $\tau_1$ and $I_1$ increased from 0.152 ns and 26.3% to 0.169 ns and 31.1% after SGF-RM, respectively. This suggests the creation of more point defects, including anti-site defects and dislocations that were introduced into the matrix after SGF-RM. The second-lifetime component $\tau_2$ was much longer than $\tau_1$ due to positron trapping and

annihilation at several large vacancy clusters or micropores. After SGF-RM, the lifetime $\tau_2$ increased, but $I_2$ decreased. The increased $\tau_2$ indicated that new micropores with larger sizes were formed in the matrix, in addition to the vacancy clusters after SGF-RM. The reduced $I_2$ may have been related to the decreased density of vacancy clusters caused by the production of more anti-site defects after tellurium evaporation. Based on the positron annihilation lifetime spectra (PALS) at different locations (top, middle, bottom) along the gravity-field direction for BST-1-R sample with the same fabrication process as BST-1 (Fig. S12 and Table 2), $\tau_1$ and $I_1$ values did not show obvious difference, showing microstructural defects distributed uniformly along the gravity-field direction for BST-1-R sample (sample of different batches samples with the same fabrication process as BST-1). The second-lifetime component $\tau_2$ was much longer than $\tau_1$ due to positron trapping and annihilation at several large vacancy clusters or micropores. The lifetime $\tau_2$ at top section are higher than that at bottom section, while the corresponding intensities $I_2$ are lower than that at bottom section, which indicated that micro-pores at top section showed larger sizes but lower density. In general, the distribution of the larger pores and microstructural defects are uniform in the matrix, while the size and density of micro-pores show a little inhomogeneous along the gravity-field direction. It is worth noting that this kind of microstructural feature did not obviously affect the thermoelectric properties. The thermoelectric properties of many samples showed good reproducibility (Figs. S7 and S8). In short, positron annihilation measurements showed that more anti-site defects, dislocations, and micropores were introduced in the $(Bi,Sb)_2Te_3$ alloys after SGF-RM.

To analyze the reasons for the reduced thermal conductivity, the microstructure and morphology were further investigated by transmission electron microscopy (TEM). No obvious dislocations were observed in the raw BST samples (Fig. S13), while high-density dislocations were found throughout the melted BST-1 samples (Figs. 3 and S14). The dislocations had diverse morphologies, with many disordered long dislocation lines existing alone or intertwining with each other to form dislocation networks (Fig. 3b). There were also many shorter dislocation lines inside the grains (Fig. S14). Most dislocations were found inside the grains instead of at the grain boundaries. The $(Bi,Sb)_2Te_3$ melt quickly cooled and solidified under supergravity, which introduced a rapid change in volume and strained the inside of the samples. This caused plastic deformation within the sample, especially inside the grains, leading to the formation of dislocation pile-ups[36,58–60].

The high-resolution TEM (HRTEM) image (Fig. 3b) shows the corresponding fast Fourier transform (FFT) pattern (inset of Fig. 3b) in the $[\overline{5}5\overline{1}]$ direction. The atomically resolved scanning transmission electron microscopy high-angle annular dark field image (Fig. 3c) shows a dislocation in the BST-1 sample. To further investigate the dislocation characteristics, inverse fast Fourier transform (IFFT) images (Fig. 3d–f) combined with geometric phase analysis (GPA) were introduced to analyze the HRTEM images for dislocation cores and corresponding strain fields (Fig. 3g–i). Many dislocation cores existed around dislocations in all three planes. The dislocation density was estimated to be $7 \times 10^{12}\,cm^{-2}$. Strain convergence regions were found around the dislocation cores, which were randomly distributed in all orientations. The high-density dislocations and the associated strain field strongly interfered with the propagation of short/medium-wavelength phonons and in the softening of the lattice[61,62], which reduced the thermal conductivity. For comparison, the IFFT images and GPA of the raw BST sample were shown in Figs. S13d–f and 3j–k, respectively. Clearly, the dislocation density in BST ($\approx 10^5$–$10^6\,cm^{-2}$) is lower than that of BST-1.

More micropores and finer grains were observed on the surfaces of the melted BST-1 samples (Fig. S15), which also confirmed the above positron annihilation measurements. From Fig. S15d–f, the pores distribution, chemical composition, and the measured density did not show obvious difference along the gravity-field direction for both BST and BST-1 samples, showing good microstructural and compositional uniformity along the gravity-field direction. These micropores and grain boundaries also helped decrease the thermal conductivity. Analogous results were also reported in previous studies[6,63]. The formation of these micropores and smaller grains was also related to the rapid solidification of the melt under the super-gravity field, as discussed above.

Based on the above positron annihilation measurements and microstructural characterization results, the presence of pile-ups of microstructural defects, such as dislocations, anti-site defects $(Bi(Sb)'_{Te})$, and micropores, was confirmed, which showed that the microstructures of the $(Bi,Sb)_2Te_3$ alloys were reconstructed after SGF-RM. The enhanced microstructural defects increased the phonon-scattering and carrier-scattering, resulting in a decrease of lattice thermal conductivity (Fig. 2e) and carrier mobility (Table 1). Based on the equation[64], $\sigma = ne\mu$ ($\sigma$ is the electrical conductivity, $n$ is the carrier concentration, $e$ is the electron charge, $\mu$ is the carrier mobility), two competing factors of carrier concentration ($n$) and mobility ($\mu$) determine the electrical conductivity. The enhanced microstructural defects contributed to the decrease in carrier mobility ($\mu$), while the evaporation of excess Te increased the carrier concentration ($n$). Because the enhanced carrier concentration contributed more to the electrical conductivity, the electrical conductivity increased instead of decreased after SGF-RM (Fig. 2a). To better understand the main factor responsible for the reduced lattice thermal conductivity, the effective medium theory (EMT) and the Debye–Callaway model were used to analyze the contributions of the absence of thermal conduction within the micropores and the various phonon scattering mechanisms, respectively (**Calculation of the thermal transport properties in Supporting information**, Fig. S16 and Table S3). According to the classical EMT, the lattice thermal conductivity of a fully dense material ($\kappa_{L,d}$) can be expressed by $\kappa_{L,d} = \kappa_{L,p}/(1-3\varepsilon/2)$, where $\kappa_{L,p}$ is the lattice thermal conductivity of the porous material, and $\varepsilon$ is the porosity[65]. Based on the experimental lattice thermal conductivity and porosity (Table S2), the corrected $\kappa_{L,d}$ of the corresponding dense BST-1 sample was obtained (open triangles in Fig. 1e). The corrected $\kappa_{L,d}$ can be described using the Debye–Callaway model to analyze the contributions of the various scattering mechanisms to the reduction in the thermal conductivity. The contributions of Umklapp scattering (U), normal scattering (N), and point defect (PD) scattering were accounted for using the Debye–Callaway model to fit the data of the BST sample (black line in Fig. 1e). Increases in the deviation upon increasing the temperature were due to a bipolar effect. Substantial heat was carried by mid/long-wavelength phonons, which could be scattered more effectively by the interfaces of micropores and grain boundaries. This resulted in a 14% – 19% reduction of the lattice thermal conductivity in the measured temperature range of 300–500 K (purple solid line in Fig. 1e). Furthermore, the greater reduction in the lattice thermal conductivity was attributed to the high dislocation density ($\approx 7 \times 10^{12}\,cm^{-2}$), which mainly scattered phonons in the short/mid-wavelength range. The absence of thermal conduction within the micropores also reduced the thermal conductivity based on the above EMT. The micropore structure (including micropores and their interfaces) resulted in about a 24% total reduction in the lattice thermal conductivity at 300 K. As a result, the reconstruction of the microstructures by SGF-RM resulted in an ultra-low lattice thermal conductivity.

The thermoelectric power generation is a more direct index used to further confirm the enhanced $zT$ values. Thermoelectric modules with 127 pairs of $p$-$n$ legs (inset in Fig. 4a) were fabricated to study the power generation. The measured conversion efficiency and power output of the thermoelectric module (BST-1 module in Fig. 4) together with the commercial module (BST module in Fig. 4) are shown in Fig. 4a and c. When the temperature difference across the module

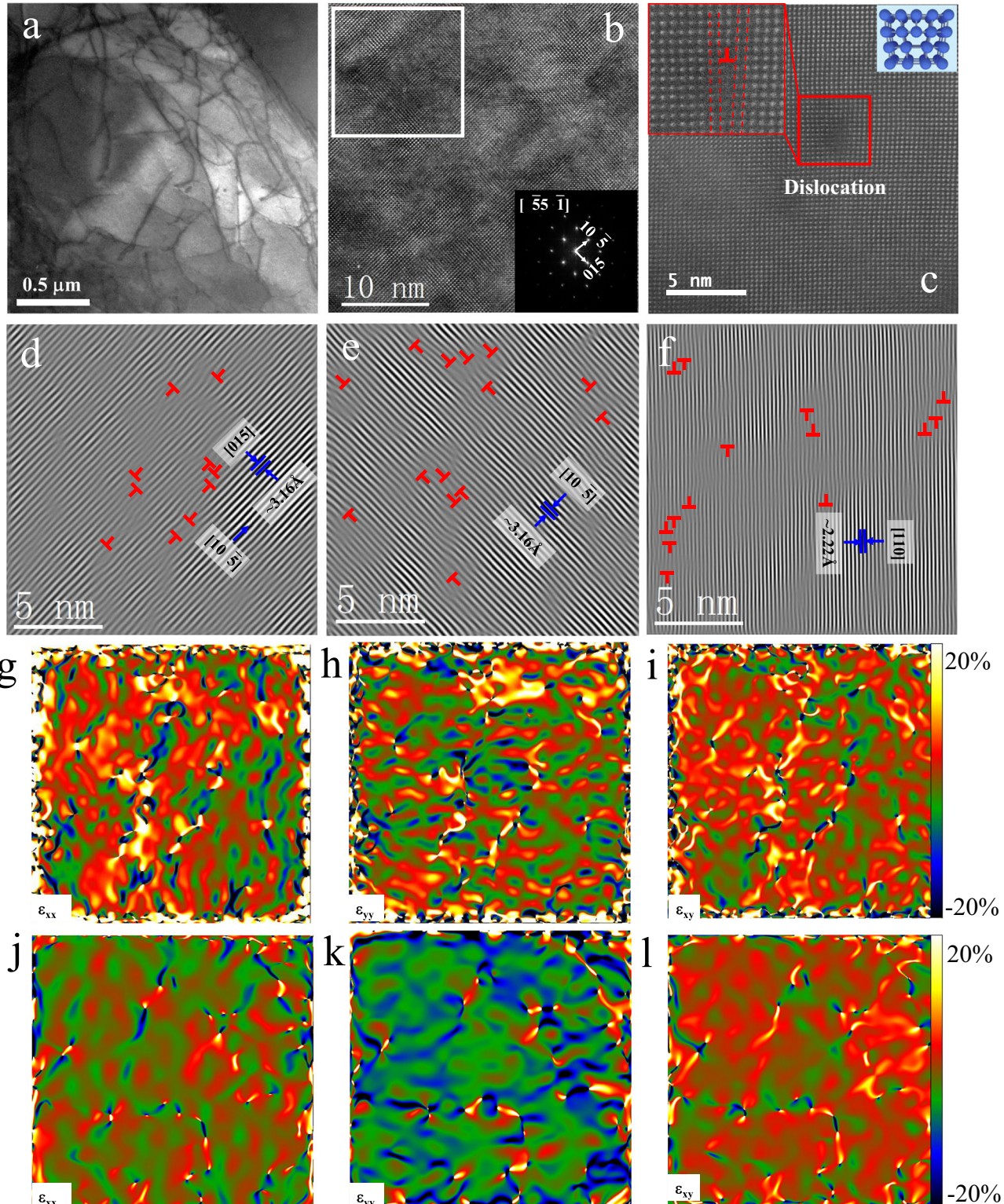

**Fig. 3 | Transmission electron microscopy (TEM) images of BST-1 sample (a–i) and BST sample (j–l). a** Low-magnification TEM Image. **b** High-resolution TEM (HRTEM) image of a randomly selected region in this figure (**a**). **c** Atomically resolved scanning transmission electron microscopy high-angle annular dark field (STEM HAADF) image showing a dislocation in a randomly selected region in this figure (**a**). **d–f** Inverse fast Fourier transform (IFFT) images in the (015), (105̄), and (110) planes obtained from the area marked by the white rectangle in Fig. 3b. **g–i** Strain field maps of $\varepsilon_{xx}$, $\varepsilon_{yy}$, and shear strain $\varepsilon_{xy}$ for BST-1 sample. **j–l** Strain field maps of $\varepsilon_{xx}$, $\varepsilon_{yy}$, and shear strain $\varepsilon_{xy}$ for BST sample. The color scale corresponds to strain from −20% to 20% with reference to the average strain of a non-defect area.

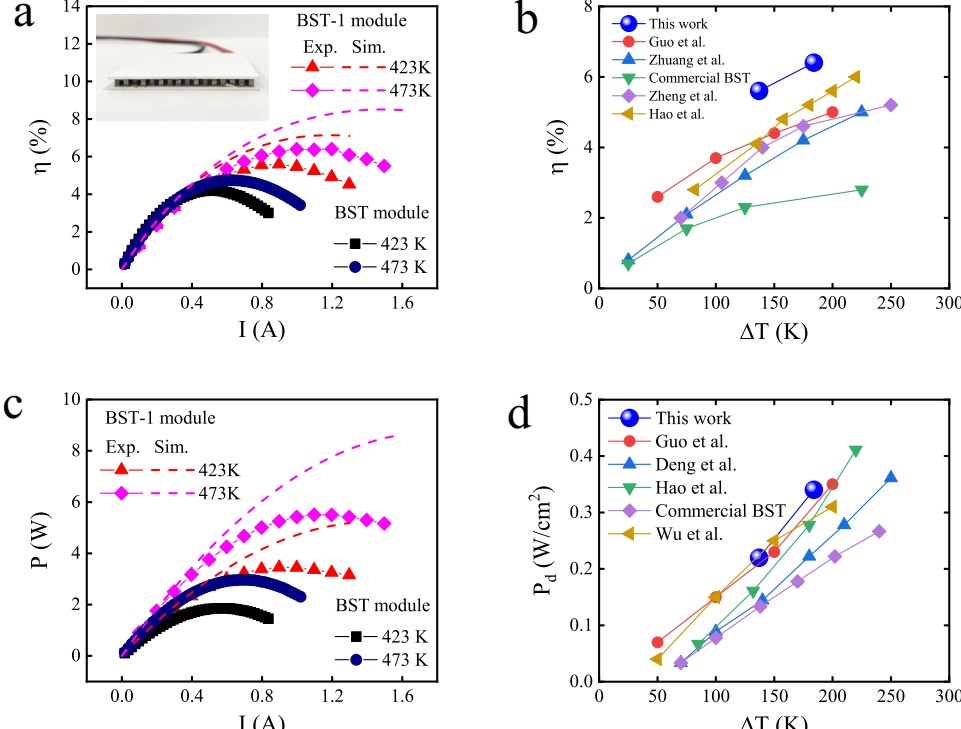

**Fig. 4 | Measured conversion efficiency and output power. a** Electric current dependence of the conversion efficiency, and the inset shows a photograph of the module. **b** Maximum conversion efficiency with different temperature differences. **c** Electric current dependence of output power. **d** Power density with different temperature differences. The hot-side temperature was set to 423 K or 473 K. The corresponding cold-side temperature was 286 K or 289 K. Data from previous studies are shown for comparison[7,40,66–69].

increased to 184 K ($T_{cold}$ = 289 K, $T_{hot}$ = 473 K), the measured maximum conversion efficiency ($\eta$) was 6.4%, which was about 52% higher than that of the commercial module ($\approx$ 4.2%). And, high output power of 5.5 W was obtained, representing 83% improvement in comparison to that of commercial module. The corresponding output power density arrived at 0.34 W/cm$^2$ (Fig. 4d). Although the conversion efficiency and power output the thermoelectric module were lower than the theoretical values, they were higher than many reported values of (Bi,Sb)$_2$Te$_3$-based devices[7,66–69] (Fig. 4b and d). These results confirmed the greatly enhanced $zT$ values of the melted (Bi,Sb)$_2$Te$_3$ alloys under super-gravity. The thermoelectric properties of the $n$-type legs (Table S4) remained much lower than those of the $p$-type legs. If the thermoelectric properties of the $n$-type material were improved and the bonding technology of the module were optimized, a higher conversion efficiency might be obtained.

The present study highlights the ultrahigh thermoelectricity in the classical BiSbTe alloy obtained by a new SGF-RM fabrication technology. Under a super-gravity field, the brittle (Bi,Sb)$_2$Te$_3$ alloy underwent unusual plastic deformation during melt solidification, which reconstructed its microstructure and formed multiple microstructure defects. Together with the carrier concentration optimization, an ultra-low thermal conductivity and a record-high figure of merit ($zT$ > 1.91 at 375 K) were obtained in the BiSbTe alloy. A high-performance thermoelectric device ($\eta_{max}$ = 6.4%) further demonstrated the enhanced thermoelectric properties and the potential applications in the power generation devices.

## Methods
### Materials synthesis
The zone-melted Te-rich Bi$_{0.48}$Sb$_{1.52}$Te$_{3.03}$ ingots (BST) are used as the starting materials. These zone-melted ingots are firstly hand-milled into powders. Some powders are cold pressed into cylindrical compacts with a diameter of 25 mm (BST-1, BST-10). Other powders are sifted. The sieved products with particles between 0.6 and 1 microns are also cold pressed (BST-S). Each batch of about 0.1 $kg$ powders was cold pressed under a uniaxial pressure of 10 MPa. The compacts are loaded into quartz ampoules, which is then evacuated and sealed. As shown in Fig. S2, the mixtures of titanium and boron powders are poured into a quartz crucible with an inner diameter of 40 mm and length of 250 mm, into which the sealed (Bi,Sb)$_2$Te$_3$ specimen is placed. A tungsten coil is fixed above the top surface of the (Ti + 2B) mixtures. The quartz crucible is wrapped with carbon felt, and then is loaded into a graphite crucible. A graphite cap is used to close the quartz and graphite crucibles. The graphite crucible is also wrapped with carbon felt and placed into a steel cup, and the cup is horizontally mounted at one side of a rotator in the reaction chamber. A counterweight is mounted at the other side of the rotator to keep balance. After the reaction chamber is evacuated, the rotator is started. By the centrifugal effect, an equivalent super-gravity field ($G$) is induced by high-speed rotation. When the super-gravity reaches set values (for example, $G$ = 100 × $g$, 1000 × $g$ ($g$ = 9.8 m/s$^2$)), the top of the (Ti + 2B) mixing powders are ignited by passing an electric current of 10 A in the tungsten coil for 2 s. After being ignited, the (Ti + 2B) powders bed continues to burn in a self-sustained way with the combustion front moving from the top to the bottom of the quartz crucible. During the burning process, a large amount of heat energy is created[28], which melts the (Bi,Sb)$_2$Te$_3$ compacts. For the exothermic reaction, Ti+2B→TiB$_2$, the flame front velocity was about 15–26.6 mm/s[70], the combustion reaction lasted for tens of seconds in this work. The peak temperature of the TiB$_2$ combustion synthesis was over 1200 °C[71], while the melting temperature of Bi$_{0.48}$Sb$_{1.52}$Te$_{3.03}$ alloy was about 610 °C[72]. So, the high temperature inside of the "chemical furnace" would hold for longer time to melt the (Bi,Sb)$_2$Te$_3$ alloys with the heat insulation layer (Fig. S2). In fact, the super-gravity field of 1000 × $g$ holds for 1 min (BST-1) and 10 min (BST-10, BST-S) to study the densification

process after being ignited. The solidified $(Bi,Sb)_2Te_3$ ingots are obtained and then are taken out for later characterizations and measurements. It is worth noting that the density is too low (the relative density is only 80.81%) for the samples fabricated under the super-gravity field of $100 \times g$. So, these samlpes are not discussed in this paper.

The simple model is used to analyze the densification process of $(Bi,Sb)_2Te_3$ alloys in the melting under super-gravity field without considering the temperature gradient, compositional gradient, and melt turbulence. Supergravity can enhance the energy transfer of multiphase flow, thus strengthening the mass transfer, heat transfer, and chemical reaction processes. So, it can be recognized that the continuous alloy melt homogenized in a moment under the super-gravity field. But there are still a lot of bubbles in the melt, which determines the density of the final alloy product. Figures 1b and S3 briefly show the densification process of $(Bi,Sb)_2Te_3$ alloy. Melts and bubbles are separated during the melting, and then cooled under super-gravity.

As we know, the lifting velocity of bubbles in melts is closed related to the supergravity coefficients. According to Stokes law[31], the lifting velocity of bubbles in melts can be calculated ($V_B$):

$$V_B = \frac{2}{9}(\rho_{M,l} - \rho_{B,g})\frac{GR_B^2}{\eta_M} \qquad (1)$$

where, $\rho_{M,l}$ and $\rho_{B,g}$ are the density of melts and bubble, respectively. $G$ is the super-gravity field, $R_B$ is the radius of bubble, $\eta_M$ is the viscosity of melts.

The forces of the bubbles in melts include the super-gravity ($F_g$), buoyancy of melts ($F_b$), and viscous drag of melts ($F_v$).

The super-gravity:

$$F_g = \frac{4}{3}\pi R_B^3 \rho_{B,g} G \qquad (2)$$

The buoyancy of melts:

$$F_b = \frac{4}{3}\pi R_B^3 \rho_{M,l} G \qquad (3)$$

The viscous drag of melts:

$$F_v = 6\pi R_B^3 \eta_M V_B \qquad (4)$$

when $F_b = F_g + F_v$, the lifting velocity of bubbles reaches a stabilized value. According to the formula (2–4), the lifting velocity of bubbles at steady state ($V_B$):

$$V_B = \frac{2}{9}(\rho_{M,l} - \rho_{B,g})\frac{R_B^2 G}{\eta_M} \qquad (5)$$

Because $\rho_{B,g} \ll \rho_{M,l}$, formula (5) can be simplified as:

$$V_B = \frac{2}{9}\rho_{M,l}\frac{R_B^2 G}{\eta_M} \qquad (6)$$

According to the formula (6), the lifting velocity of bubbles in alloy melts at steady state is calculated (by using the data listed in Table 1) and shown in Fig. S3.

The above results show that the lifting velocity of bubbles in alloy melts can be obviously increased by enhancing the supergravity coefficient ($G/g$) and the radius of bubbles. For $(Bi,Sb)_2Te_3$ alloy, the super-gravity coefficient ($G/g$) of 1000 is high enough to densify the bulks. However, the bubbles with small radii are hard to clean out of the alloy melts in the short densification process due to the lower

lifting velocity of small bubbles. So, a few small pores are observed in the melted bulks.

## Structural characterization

The phase composition is analyzed by X-ray diffraction (Bruker, Germany) with Cu Kα radiation. The typical XRD patterns of the $(Bi,Sb)_2Te_3$ samples are shown in Fig. S4. The microstructures are observed by field-emission scanning electron microscopy (S-4800, Hitachi) and TEM (2100F, JEOL). Elemental analyses are collected by inductively coupled plasma-optical emission spectroscopy (ICP-OES, Varian 710-ES).

## Thermoelectric property measurements

The Seebeck coefficient ($\alpha$) and electrical conductivity ($\sigma$) are measured by using the Seebeck Coefficient/Electrical Resistance Measuring System (ZEM-3, Ulvac-Riko) under a static helium atmosphere. The Hall coefficient ($R_H$) is measured by a Hall measurement system (ResiTest 8340DC, Toyo, Japan) via the van der Pauw method. The hall carrier concentration ($n_H$) and mobility ($\mu_H$) are calculated by $n_H = 1/(eR_H)$ and $\mu_H = R_H/\rho$, respectively. The thermal conductivity ($\kappa$) is calculated using the equation $\kappa = \lambda C_p d$, where $\lambda$ is the thermal diffusivity, $C_p$ is the heat capacity, and $d$ is bulk density of the sample. The thermal diffusivity is measured by a laser flash technique (Netzsch LFA457) in $Ar$ atmosphere. The heat capacity is measured using Differential Scanning Calorimeter (DSC404-F3). The measured $\lambda$ and $C_p$ values were shown in Figs. S5 and S6, respectively. The bulk density is obtained by the Archimedes method. The lattice thermal conductivities ($\kappa_L$) are obtained by subtracting the electrical contribution from the total thermal conductivity using the equation $\kappa_L = \kappa \cdot \kappa_e$. Here, the electrical thermal conductivity is expressed by the Wiedemann–Franz Law $\kappa_e = L\sigma T$, where $L$ is estimated by using a Single Parabolic Band (SPB) model[73]. Transport properties are measured in the parallel (cross-plane) (Fig. 2) and perpendicular (in-plane) (Fig. S9) to the direction of the super-gravity field. The transport properties in the parallel direction were repeated (Figs. S7 and S8). Usually, the uncertainties of commercial instruments are ±4% for α, ± 3% for σ, ± 3% for λ, ± 5% for $C_p$[74]. The measured error of Archimedes method and thermocouple temperature measurement are very small. So, the measured error from $d$ and $T$ can be neglected. Combining the electrical conductivity, Seebeck coefficient, and thermal conductivity obtained from the measurements, the estimated accumulated measured error (uncertainty) of $zT$ is about ±15%.

## Positron annihilation measurement

Positron annihilation lifetime spectroscopy (PALS) analysis is performed using a fast-slow coincident ORTEC system with a time resolution of 220 $ps$ for the full width at half maximum. The $^{22}$Na positron source is placed between the two pieces of samples, and then the "sample-source-sample sandwich" is placed between the two $BaF_2$ detectors to acquire the lifetime spectra. A total of $2 \times 10^6$ counts are accumulated for each spectrum to reduce the statistical error in the calculation of lifetimes. The positron lifetime spectra are de-convoluted and analyzed using the LT-9 software. LT-9 is one of the most popular software for PALS analysis. It de-convolutes the experimental curve from the instrument functions to set apart the physical meaning information, i.e., positron annihilation lifetime and intensity. Positron annihilation lifetime and intensity could reflect the defect size and density information.

## Module fabrication and measurement

TE modules with the size of $40 \times 40 \times 2.6$ mm$^3$ and a total 127 pairs of $p$-$n$ legs were fabricated in Huabei Cooling Device Company. The size for the legs is $1.33 \times 1.33 \times 1.6$ mm$^3$. The melted $Bi_{0.48}Sb_{1.52}Te_{3.03}$ sample was utilized for the $p$-type legs. The zone-melted BiSbTe alloys (BST) serve

as the references. The *n*-type counterparts are commercial $Bi_2Te_{2.2}Se_{0.8}$ ingots (The measured thermoelectric parameters were shown in Table S4). The energy conversion efficiencies and cooling temperature difference of these modules were evaluated by man-made testing system in Shenzhen Institute of Advanced Technology, Chinese Academy of Sciences. The hot-side temperatures were maintained between 423 and 500 K, while the temperature of the water cooler was kept at 283 K.

## Data availability

All data necessary to understand and assess this manuscript are shown in the main text and the Supporting Information. The data that support the findings of this study are available from the corresponding author on request.

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

## Acknowledgements

This work was supported by the Key Research Program of the State Key Laboratory of Cryogenic Science and Technology (CRYO20230203, awarded to M.Z.), the National Natural Science Foundation of China (51872299, awarded to M.Z.), and the Basic Science Center Project of National Natural Science Foundation of China (52388201, awarded to M.Z.). We thank Tsinghua University, Beihang University, and Ningbo Institute of Materials Technology & Engineering, CAS for the repeated measurements of the thermoelectric properties. We also thank Huabei Cooling Device Co., Ltd. for fabricating the thermoelectric device modules and the Shenzhen Institute of Advanced Electronic Materials for measuring the thermoelectric conversion efficiency.

## Author contributions

M.Z. and H.J.S. synthesized the samples, designed and carried out the experiments, analyzed the results, and wrote the paper. J.P. conducted theoretical calculations and analyzed the results. L.W. measured the thermoelectric performance. H.L.Z. measured the Hall coefficient. K.S., H.Y.H., and J.J. fabricated the thermoelectric modules. Q.H.Z. helped with TEM measurements and analysis. J.T.L. guided the super-gravity-field re-melting process. The experimental design and paper writing were performed under the supervision of J.-F.L., J.J., and L.F.L. All authors contributed to the discussion of the results and commented on the manuscript.

## Competing interests

The authors declare no competing interests.
