## [Transparent Peer Review file · Nature Communications]

Ultrahigh thermoelectricity obtained in classical BiSbTe alloy processed under super-gravity

Corresponding Author: Professor Min Zhou

Version 0:

Reviewer comments:

Reviewer #1

(Remarks to the Author)

This manuscript presents an innovative super-gravity-field remelting process that demonstrates clear novelty in thermoelectric materials processing. The authors present their concept in a well-organized manner, and the study effectively establishes the relationship between optimized carrier concentration, resulting microstructure, and enhanced thermoelectric properties of the fabricated BST.

The improvements in thermoelectric performance are well-justified and supported by the experimental results. The manuscript is logically structured with consistent presentation throughout.

The authors have successfully demonstrated the process-structure-property relationships in their material system. While the overall quality of the work is high, several points need to be addressed before acceptance. Please refer to the specific comments below for these revisions.

My comments are as follows.

1. A reference citation is needed for the amorphous limit mentioned in lines 153-154 on page 8.
2. The significant enhancement in electrical conductivity despite the formation of finer grains, dislocations and micropores after the SGF process requires further clarification. To better understand this phenomenon, please provide Hall measurement results for all samples as a Table or Figure.
3. Include a detailed discussion analyzing how the microstructural changes (finer grains and micropores) correlate with the electrical transport properties based on the Hall measurement data.
4. The statement in lines 228-229 on page 12 need to be supported by appropriate literature citations.
5. To enable a clear comparison of the strain field, Figure 3 should include strain field images of the BST sample before SGF alongside the current images.
6. Direct evidence for the claimed micropore structure is missing from the manuscript. The fractured surface images provided in Figure S12 are insufficient to support. Please replace or supplement these with SEM images of well-polished surfaces.
7. In Figure S8, the identity of the four blue symbols needs clarification. If these data points represent different samples, use distinct symbols/colors for each of the four samples to clearly differentiate them or use different symbols to distinguish between before and after SGF-RM process.

Reviewer #2

(Remarks to the Author)

The authors synthesize BiSbTe alloy under super gravity to improve its thermoelectric performance. The manuscript needs further revision before acceptance.

The authors mention few approaches adopted to improve the thermoelectric performance of materials. It is essential to mention some of the important approaches like perfect convergence of bands, hyperconvergence of bands, introduction of multiple valleys, Rashba splitting etc with respective examples adopted to improve the performance of telluride-based materials in the introduction section.

Fig S3 could be further improved by increasing the line thickness.

The trend in the values of Seebeck for various samples change after 400 K. Further explanation should be provided why different samples show different trends with variation in the temperature.

The band gap of the synthesized materials should be determined.
Values of carrier concentration and mobility should be provided in the main article.
The zT of the material should be compared with other classes of materials containing Bi, Sb and Te.
The image (photos) of the experimental setup could be provided in the supporting information.
The zT average values should be compared with other previously reported materials.

Reviewer #3

(Remarks to the Author)

The paper presents a novel process for fabricating the BiSbTe (BST) material, a classical thermoelectric system. It is reported that the BST sintered under a super-gravity condition achieves an exceptionally high zT up to 1.91 that exceeds the value of the hot-pressed nanostructured BST (zT~1.4). This claim represents a significant advancement in thermoelectric material development. A careful examination of the fabrication process and characterization details is essential. Here are the inquiries and comments regarding the paper.

1. In a super-gravity and high-temperature firing process, controlling microstructural and compositional uniformity is crucial. This information regarding the mass density, pore distribution, and chemical composition can be obtained by sampling the material at various locations along the gravity-field direction.
2. The reproducibility testing was done on the materials fabricated by using different powder size (BST-S) and firing durations (BST-1 & BST-10). I thought the reproducibility test should be done on multiple samples prepared with the same process. Although the thermoelectric properties of the BST-S, BST-1, and BST-10 look comparable in Fig. S6 due to plot scaling, the variations in each property indeed are significant. A thorough analysis of the experimental variations, including the sampling plan, should be provided.
3. The compositional variation among the BST-S, BST-1, and BST-10 sample is quite significant, as listed in Table S2. It seems to me the raw powder size plays a role during the firing process. The BST-S with a small powder size received more loss in Te composition.
4. The BST materials after the SGF-RM process appear to have a much lower mass density (~87% of theoretical density). It definitely contributes to a low thermal conductivity. I am curious why the BST-1 exhibits a much higher electrical conductivity than the BST (98% of theoretical density). A physical explanation is necessary.
5. How is the distribution of micro-pores inside the BST materials after SGF-RM process? Is it possible to have a gradient distribution along the gravity direction, as shown in Fig. 1B. If so, it may affect the thermoelectric properties of the samples. It is suggested to perform microstructural and PALS measurement at different of the material along the gravity-field direction.
6. The dislocation density of BST-1 was estimated to be $7 \times 10^{12} \text{ cm}^{-2}$. Please provide the dislocation density of the original BST sample for comparison.

In summary, the results presented in this paper are impressive. However, the zT values were determined from individual measurements of transport properties, which may introduce some variations. Therefore, a detailed analysis of the experimental measurements is necessary to substantiate any claims of a record-high zT value.

Version 1:

Reviewer comments:

Reviewer #2

(Remarks to the Author)

The authors have revised the manuscript considerably and can be accepted in the current form.

Reviewer #3

(Remarks to the Author)

I think the revision has partially addressed some of my considerations on the original manuscript regarding the uniformity of material properties. However, some points still require further clarification.

1. The authors prepared multiple BST-1-R samples to test the reproducibility of the process proposed. It is claimed that the standard error of zT values is less than 5%. Since the zT is calculated from the individual transport property measurements, it is difficult to be convinced that the accumulated errors in zT are less than 5%. A rigorous error estimation based on the respective properties should be provided.
2. In the rebuttal letter, the authors noted that the quartz tubes containing the small particle samples (BST-S) broke during the preparation process due to the higher Te vapor pressure. The vigorous evaporation of Te may explain the significantly lower Te content in the BST-S compared to the BST-1 and BST-10 samples. As proposed by the authors, excess Te evaporation facilitates the formation of antisite defects and increases the carrier concentration of the BST-based samples. It is thus expected that the BST-S sample should exhibit a higher carrier concentration and a lower Seebeck coefficient. This expectation seems to align with the results shown in Fig. 2, but is contradictory to the Hall measurement results presented in Table 1.
3. It is curious to me why the BST-S, with such a high Te content deviation, still possesses comparable transport properties compared to the BST-1 and BST-10 samples.

Version 2:

Reviewer comments:

Reviewer #3

(Remarks to the Author)

I believe my comments and suggestions have been adequately addressed in the revised manuscript and the rebuttal letter. I am pleased to recommend the publication of this paper.

Dear Reviewers,

Many thanks for your report on our manuscript entitled: “Ultrahigh thermoelectricity obtained in classical BiSbTe alloy processed under super-gravity” (Manuscript ID: NCOMMS-25-09159-T). We really appreciate the professional comments. Our point-to-point responses for all reviewers are listed as below:

Reviewer #1 (Remarks to the Author):

This manuscript presents an innovative super-gravity-field remelting process that demonstrates clear novelty in thermoelectric materials processing. The authors present their concept in a well-organized manner, and the study effectively establishes the relationship between optimized carrier concentration, resulting microstructure, and enhanced thermoelectric properties of the fabricated BST. The improvements in thermoelectric performance are well-justified and supported by the experimental results. The manuscript is logically structured with consistent presentation throughout. The authors have successfully demonstrated the process-structure-property relationships in their material system. While the overall quality of the work is high, several points need to be addressed before acceptance. Please refer to the specific comments below for these revisions.

My comments are as follows.

1. A reference citation is needed for the amorphous limit mentioned in lines 153-154 on page 8.

Response: Thanks for your kind suggestion. The phrase “**calculated by the Cahill model^{47,48}**” was added at the end of the sentence “An ultra-low lattice thermal conductivity of 0.15–0.25 W/m K was obtained, which approached the amorphous limit” in lines 153-154 on page 8. All the references were updated.

47. Cahill, D. G., Watson, S. K. & Pohl, R. O. Lower limit to the thermal-conductivity of disordered crystals. *Phys. Rev. B* **46**, 6131-6140 (1992).

48. Dong, J. et al. Reducing Lattice Thermal Conductivity of MnTe by Se Alloying toward High Thermoelectric Performance. *ACS Appl. Mater. Inter.* **11**, 28221-28227 (2019).

2. The significant enhancement in electrical conductivity despite the formation of finer grains, dislocations and micropores after the SGF process requires further clarification. To better understand this phenomenon, please provide Hall measurement results for all samples as a Table or Figure.

Response: Yes, we measured the Hall coefficient and showed the results in the raw supporting information. In order to better support the electrical transport properties in the main text, we showed the Hall measurement in Table 1 behind the line 1 on page 9 in the revised version. The measured Hall coefficient and electrical resistivity were added in Table 1. The serial number of all the tables has been updated accordingly. The further discussion about electrical conductivity was shown in the answer of the next question.

Table 1 The ICP-OES results, the corresponding Hall measurement of all the samples before and after SGF-RM.

samples	ICP-OES results	Carrier concentration (10^{19} cm^{-3})	Mobility ($\text{cm}^2/\text{V S}$)	Hall coefficient (cm^3/C)	Electrical resistivity ($10^{-3} \Omega \text{ cm}$)
BST	$\text{Bi}_{8.51}\text{Sb}_{28.92}\text{Te}_{62.57}$	1.37	388.98	0.46	1.17
BST-S	$\text{Bi}_{9.69}\text{Sb}_{30.31}\text{Te}_{56.99}$	2.08	279.01	0.30	1.03
BST-1	$\text{Bi}_{10.56}\text{Sb}_{29.44}\text{Te}_{62.17}$	2.42	358.62	0.26	0.72
BST-10	$\text{Bi}_{8.57}\text{Sb}_{29.12}\text{Te}_{62.30}$	2.69	315.06	0.23	0.74

3. Include a detailed discussion analyzing how the microstructural changes (finer grains and micropores) correlate with the electrical transport properties based on the Hall measurement data.

Response: The further discussion about electrical transport properties is as follows:

The phrase “which was related to the increased carrier concentration (Table 1)” was added behind the sentence “After SGF-RM, the electrical conductivity increased but the Seebeck coefficient decreased” in line 21 on page 7. The sentence “The peak Seebeck coefficient decreased and moved to a higher temperature after SGF-RM” in lines 21-22 on page 7 was replaced by “Besides, the increased carrier concentration inhibited the intrinsic excitation^{45,46} and pushed the peak Seebeck coefficient to higher temperatures (Fig. 2B). So, the trend of Seebeck coefficient changed at higher temperatures after SGF-RM.”. And the sentences “These negatively charged anti-site defects formed extra holes in the matrix, which increased the carrier concentration and electrical conductivity after SGF-RM. Moreover, the maximum Seebeck coefficient shifted to a higher temperature after SGF-RM (Fig. 2B), which was also ascribed to the suppression of the intrinsic conductivity by increasing the carrier

concentration. The carrier mobility decreased after SGF-RM, primarily due to increased carrier–carrier scattering and enhanced point defect scattering.” in lines 15-18 on page 9 were replaced by “These negatively charged anti-site defects formed extra holes in the matrix, which increased the carrier concentration (**Table 1**). As a result, the electrical conductivity increased but the Seebeck coefficient decreased after SGF-RM. The carrier mobility decreased after SGF-RM, primarily due to increased carrier–carrier scattering and enhanced microstructural defects scattering, which would be discussed in the following. A little Te volatilization after SGF-RM resulted in increased carrier concentration but did not obviously change the band structure of the BiSbTe compound. The measured band gap showed consistency before and after SGF-RM (**Fig. S10**).” The sentences “The enhanced microstructural defects increased the phonon-scattering and carrier-scattering, resulting in the decrease of lattice thermal conductivity (**Fig. 2e**) and carrier mobility (**Table 1**). Based on the equation⁶⁴, $\sigma = ne\mu$ (σ is the electrical conductivity, n is the carrier concentration, e is the electron charge, μ is the carrier mobility), two competing factors of carrier concentration (n) and mobility (μ) determine the electrical conductivity. The enhanced microstructural defects contributed to the decrease in carrier mobility (μ), while the evaporation of excess Te increased the carrier concentration (n). Because the enhanced carrier concentration contributed more to the electrical conductivity, the electrical conductivity increased instead of decreased after SGF-RM (**Fig. 2a**).” was added before the sentence “To better understand the main factor responsible for the reduced lattice thermal conductivity, the effective medium theory (EMT) and the Debye–Callaway model were used to analyze the contributions of the absence of thermal conduction within the micropores and the various phonon scattering mechanisms, respectively.” in lines 10-13 on page 14.

4. The statement in lines 228-229 on page 12 need to be supported by appropriate literature citations. Response: Four related references^{36,58-60} were added at the end of the sentence in lines 228-229 on page 12.

36. Liu, G. et al. Direct fabrication of highly-dense Cu₂ZnSnSe₄ bulk materials by combustion synthesis for enhanced thermoelectric properties. *Mater. Design* **93**, 238-246 (2016).

58. Davoudi, K. M. & Vlassak, J. J. Dislocation evolution during plastic deformation: Equations vs. discrete dislocation dynamics study. *J. Appl. Phys.* **123**, 085302 (2018).

59. Messerschmidt, U. & Bartsch, M. Generation of dislocations during plastic deformation. *Mater.*

Chem. Phys. **81**, 518-523 (2003).

60. Bennett, D. C. & Sawyer, B. Single crystals of exceptional perfection and uniformity by zone leveling. Bell System Technical Journal **35**, 637-660 (1956).

5. To enable a clear comparison of the strain field, Figure 3 should include strain field images of the BST sample before SGF alongside the current images.

Response: The strain field images of the BST sample before SGF were shown in **Fig. 3j-i**. The inverse fast Fourier transform (IFFT) images were shown in **Fig. S13d-f**. The sentence “For comparison, the inverse fast Fourier transform (IFFT) images and geometric phase analysis (GPA) of the raw BST sample were shown in **Fig. S13d-f** and **Fig. 3j-k**, respectively. Clearly, the dislocation density in BST ($\approx 10^5 - 10^6 \text{ cm}^{-2}$) is lower than that of BST-1.” was added at the end of the second paragraph on page 12.

Fig. 3. Transmission electron microscopy (TEM) images of **BST-1 sample (a-i) and **BST sample (j-l)**.** (a) Low-magnification TEM Image. (b) High-resolution TEM (HRTEM) image of a randomly selected region in **Fig. 3a**. (c) Atomically resolved scanning transmission electron microscopy high-angle annular dark field (STEM HAADF) image showing a dislocation in a randomly selected region in **Fig. 3a**. (d–f) Inverse fast Fourier transform (IFFT) images in the (015), (10 $\bar{5}$) and (110) planes obtained from the area marked by the white rectangle in **Fig. 3b**. (g–i) Strain field maps of ϵ_{xx} , ϵ_{yy} , and shear strain ϵ_{xy} for **BST-1 sample**. (j–l) Strain field maps of ϵ_{xx} , ϵ_{yy} , and shear strain ϵ_{xy} for **BST sample**. The color scale corresponds to strain from -20% to 20% with reference to the average strain

of a non-defect area.

Fig. S13 TEM images of the BST sample. (a, b) Low-magnification in different regions, (c) high-magnification, (d–f) Inverse fast Fourier transform (IFFT) images in the (110), (2 $\bar{1}0$) and ($\bar{1}20$) planes obtained from the area marked by the white rectangle in Fig. S12c.

6. Direct evidence for the claimed micropore structure is missing from the manuscript. The fractured surface images provided in Figure S13 are insufficient to support. Please replace or supplement these with SEM images of well-polished surfaces.

Response: Some typical polished surface SEM images of BST and BST-1 were supplied. The corresponding chemical compositions and measured density at different locations (top, middle, bottom) along the gravity-field direction were also shown. Figure S13 was updated to Fig. S15 in the revised version.

Fig. S15 The typical SEM images, chemical compositions and measured density at different locations (top, middle, bottom) along the gravity-field direction for BST and BST-1 samples. (a) The fractured surface SEM images of BST (lower magnification), (b) The fractured surface SEM images of BST-1 (lower magnification), (c) The fractured surface SEM images of BST-1 (higher magnification), (d) The polished surface SEM images and EDS results of BST, (e) The polished surface SEM images and EDS results of BST-1, (f) Measured density at different locations for BST and BST-1. The chemical compositions and density did not show obvious difference along the gravity-field direction.

7. In Figure S8, the identity of the four blue symbols needs clarification. If these data points represent different samples, use distinct symbols/colors for each of the four samples to clearly differentiate them or use different symbols to distinguish between before and after SGF-RM process.

Response: The four blue symbols represented four different samples and were shown by different colors. Figure S8 was updated to Fig. S11 in the revised version.

Fig. S11 The charge carrier transport properties. (a) Hall carrier concentration dependence of Seebeck coefficient (Pisarenko plot) at 300 K, (b) Hall carrier concentration dependence of Hall mobility at 300 K, (c) Electrical conductivity dependence of Seebeck coefficient at 300 K, (d) Power factor (PF) as a function of Hall carrier concentration predicted by $m^*=1.05 m_0$ and $\mu_0=420 \text{ cm}^2/\text{Vs}$ at 300 K. The bigger filled circles represent the data of this work. The other smaller filled symbols represent literature data^{6,7,27,36,40,48}.

Reviewer #2 (Remarks to the Author):

The authors synthesize BiSbTe alloy under super gravity to improve its thermoelectric performance. The manuscript needs further revision before acceptance.

1. The authors mention few approaches adopted to improve the thermoelectric performance of materials. It is essential to mention some of the important approaches like perfect convergence of bands, hyperconvergence of bands, introduction of multiple valleys, Rashba splitting etc. with respective examples adopted to improve the performance of telluride-based materials in the introduction section.

Response: Thanks for your kind suggestion. We updated the introduction section. The sentence “ Bi_2Te_3

is a remarkably good thermoelectric material naturally having low lattice thermal conductivity and complex electronic structure.” was added after the sentence “Although a lot of new thermoelectric materials have been discovered and received more attention, Bi₂Te₃-based alloys remain at the forefront of thermoelectric research” in lines 16-18 on page 3. In line 23 on page 3 and line 1 on page 4, the sentences “but the zT value of higher than unity has rarely been obtained. Until 2008, a higher zT of 1.4 was reported¹, which stimulated a series of following studies.” were deleted. The sentence “A higher zT of 1.4 (373 K) was reported in nanocrystalline BiSbTe alloy¹⁹.” was added before the sentence “However, no significant increase in zT values was realized until *Science* published another work, which obtained a high zT of 1.86 in (Bi,Sb)₂Te₃ compounds via liquid phase sintering with excess *Te*” in lines 5-7 on page 5.

The sentences “The band structure engineering is an effective approach in the enhancement of thermoelectric performance for Bi₂Te₃. Solid solution alloying with Sb₂Te₃ or Bi₂Se₃ induces band convergence which increases the density of states (DOS) as well as reduces lattice thermal conductivity¹¹. In the p-type (Bi,Sb)₂Te₃ system, valence band convergence occurs near the most commonly used composition Bi_{0.5}Sb_{1.5}Te₃ and the corresponding peak zT value is about 1 (300K)¹². *Sn* impurity in the valence band of Bi₂Te₃ enhances Seebeck coefficient through resonant scattering^{13,14}. *Au* doping on the *Bi* site of Bi₂Te_{2.7}Se_{0.3} also induces resonant states, leading to increase of the Seebeck coefficient. Peak zT value of 0.91 is obtained for Cu_{0.008}Bi_{1.99}Au_{0.01}Te_{2.7}Se_{0.3} (320K)¹⁵. CuI-doped (CuI)_xBi₂Te_{2.7}Se_{0.3} increases the crystalline electric field, which results in the Rashba band splitting. The formation of Rashba band effect enhances the power factor and zT value in a wide temperature range^{16,17}. The point defect engineering is also effective to optimize thermoelectric properties of Bi₂Te₃-based alloys. Antisite defects (Bi_{Te}’, Sb_{Te}’) and donor-like effects are engineered by tuning the formation energy of point defects¹⁸.” were added before the sentence “Recent studies have focused on investigating structural modification to enhance the zT values of polycrystalline Bi₂Te₃ alloys...” in lines 1-5 on page 4.

11. Witting, I. T. et al. The Thermoelectric Properties of Bismuth Telluride. *Advanced Electronic Materials* **5**, 201800904 (2019).

12. Kim, H.-S. et al. High thermoelectric performance in (Bi_{0.25}Sb_{0.75})₂Te₃ due to band convergence and improved by carrier concentration control. *Mater. Today* **20**, 452-459 (2017).

13. Jaworski, C. M., Kulbachinskii, V. & Heremans, J. P. Resonant level formed by tin in Bi₂Te₃ and

the enhancement of room-temperature thermoelectric power. *Phys. Rev. B* **80**, 233201 (2009).

14. Heremans, J. P., Wiendlocha, B. & Chamoire, A. M. Resonant levels in bulk thermoelectric semiconductors. *Energ. Environ. Sci.* **5**, 5510-5530 (2012).

15. Lee, K. H. et al. Enhanced thermoelectric performance of n-type $\text{Cu}_{0.008}\text{Bi}_2\text{Te}_{2.7}\text{Se}_{0.3}$ by band engineering. *J. Mater. Chem. C* **3**, 10604-10609 (2015).

16. Kim, J. H. et al. Possible Rashba band splitting and thermoelectric properties in CuI-doped $\text{Bi}_2\text{Te}_{2.7}\text{Se}_{0.3}$ bulk crystals. *J. Alloy Compd.* **806**, 636-642 (2019).

17. Ishizaka, K. et al. Giant Rashba-type spin splitting in bulk BiTeI . *Nat. Mater.* **10**, 521-526 (2011).

18. Hu, L., Zhu, T., Liu, X. & Zhao, X. Point Defect Engineering of High-Performance Bismuth-Telluride-Based Thermoelectric Materials. *Adv. Funct. Mater.* **24**, 5211-5218 (2014).

2. Fig S3 could be further improved by increasing the line thickness.

Response: The lines in Fig. S3 were bolded. The Fig. S3 was updated to Fig. S4.

Fig. S4 The powder X-ray diffraction patterns of all the samples before and after SGF-RM.

3. The trend in the values of Seebeck for various samples change after 400 K. Further explanation should be provided why different samples show different trends with variation in the temperature.

Response: In lines 20-22 on page 7, the sentences “After SGF-RM, the electrical conductivity increased but the Seebeck coefficient decreased. The peak Seebeck coefficient decreased and moved to a higher temperature after SGF-RM.” were deleted. The sentences “which was related to the

increased carrier concentration (**Table 1**). Besides, the increased carrier concentration inhibited the intrinsic excitation^{45,46} and pushed the peak Seebeck coefficient to higher temperatures (Fig. 2b). So, the trend of Seebeck coefficient changed at higher temperatures after SGF-RM.” were added after the sentence “After SGF-RM, the electrical conductivity increased but the Seebeck coefficient decreased” in line 21 on page 7.

4. The band gap of the synthesized materials should be determined.

Response: We measured the Fourier transform infrared absorption spectrum (middle-infrared band, Thermo Fisher Nicolet iS50) and estimated the band gap values of all samples. The results were shown in Fig. S10 in the revised version. The reported band gap of $\text{Bi}_{0.5}\text{Sb}_{1.5}\text{Te}_3$ alloy is about 0.19 eV[5]. The estimated band gap values by the properly extrapolated technique are lower than the reported values. The small error in the band gap values may be related to the chemical compositions, grain sizes, porosity, fabrication process and extrapolation method etc.

Fig. S10. The optical test results. (a) Fourier transform infrared absorption spectrum (middle-infrared band) of all samples, (b) Tauc plot of all samples. The dotted lines indicate the properly extrapolated technique. It is worth noting that the traditional method of directly extrapolating the linear segment of the Tauc plot for band gap semiconductor is inappropriate for use on degenerate semiconductors, where the occupation of conduction band energy states cannot be ignored^{80, 81}. We used the properly extrapolated technique for extracting the optical band gap from absorption spectra by introducing a baseline function such that a vertical line (or ordinate) is then dropped to intersect the $h\nu$ axis (Fig. S10b)^{82,83}. The proper extrapolation band gap is about 0.168 eV for all the $(\text{Bi,Sb})_2\text{Te}_3$ samples before and after SGF-RM. The reported band gap of $\text{Bi}_{0.5}\text{Sb}_{1.5}\text{Te}_3$ alloy is about

0.19 eV⁸⁴. The estimated band gap values by the properly extrapolated technique are lower than the reported values. The small error in the band gap values may be related to the chemical compositions, grain sizes, porosity, fabrication process and extrapolation method etc.

5. Values of carrier concentration and mobility should be provided in the main article.

Response: We deleted Table S2 in the supporting information and put it after Fig. 2 on page 8. The name of the table was Table 1. The content of the table was also updated.

Table 1 The ICP-OES results, the corresponding Hall measurement of all the samples before and after SGF-RM.

samples	ICP-OES results	Carrier concentration (10 ¹⁹ cm ⁻³)	Mobility (cm ² /V S)	Hall coefficient (cm ³ /C)	Electrical resistivity (10 ⁻³ Ω cm)
BST	Bi _{8.51} Sb _{28.92} Te _{62.57}	1.37	388.98	0.46	1.17
BST-S	Bi _{9.69} Sb _{30.31} Te _{56.99}	2.08	279.01	0.30	1.03
BST-1	Bi _{10.56} Sb _{29.44} Te _{62.17}	2.42	358.62	0.26	0.72
BST-10	Bi _{8.57} Sb _{29.12} Te _{62.30}	2.69	315.06	0.23	0.74

6. The zT of the material should be compared with other classes of materials containing Bi, Sb and Te.

Response: In Fig. 1g, we added some typical zT values of (Bi,Sb)₂Te₃-based materials. Some reported typical zT average values have been shown in Fig. 2f for comparison. The sentence “the zT value reached >1.91 (375 K) for the re-melted (Bi,Sb)₂Te₃ alloy under super-gravity (Fig. 1g)” in lines 4-5 on page 6 was updated to “high zT value of over 1.91 (375 K) was obtained for the re-melted (Bi,Sb)₂Te₃ alloy under super-gravity (Fig. 1g), which was higher than many reported data^{6,7,19,27,40-42}”.

Fig. 1. Synergistically optimizing phonon and electron transport for record-high zT values.

Schematic illustration of the (a) super-gravity-field re-melting technology, (b) movement of bubbles in melts, and (c) reconstruction of microstructures after super-gravity-field re-melting (SGF-RM). (d) Process of Te evaporation causing extra holes. (e) Lattice thermal conductivities (κ_L) of samples before and after SGF-RM. The solid symbols present the experimental results. The black solid line represents the predicted κ_L value considering the scattering of the Umklapp process, normal process, and point defects (U+N+P). The purple solid line represents the predicted κ_L value considering the additional scattering of grain boundaries and micro-pore interfaces (U+N+P+I). The red solid line represents the predicted κ_L values considering the additional scattering of dislocations (U+N+P+I+DS). The effective medium theory (EMT)-corrected values are shown by red empty triangles. (f) Power factor values as a function of the Hall carrier concentration predicted by the effective mass $m^* = 1.05m_0$ and drift mobility $\mu_w = 420 \text{ cm}^2/\text{V s}$ at 300 K. (g) zT values of the $\text{Bi}_{0.48}\text{Sb}_{1.52}\text{Te}_{3.03}$ alloy before (BST) and after SGF-RM. Note: The sample with hand-milled powders is denoted as BST-1 after re-melting under super-gravity for 1 min. The sample with hand-milled powders is denoted as BST-10 after re-melting under super-gravity for 10 min. The sample with

particle sizes between 0.6 and 1 μm is denoted as BST-S after re-melting under super-gravity for 10 min. Some reported typical results of $(\text{Bi,Sb})_2\text{Te}_3$ -based materials are also shown in Fig. 1g^{6,7,19,27,40-}

42.

7. The image (photos) of the experimental setup could be provided in the supporting information.

Response: The photos of the super-gravity-field re-melting setup were show in Fig. S1 on page 9. The figure numbers were updated accordingly.

Fig. S1 The real photos of the super-gravity re-melting setup. (a) equipment appearance, (b) internal rotor.

8. The zT average values should be compared with other previously reported materials.

Response: Yes, some reported typical zT average values have been shown in Fig. 2f for comparison.

Fig. 2. Thermoelectric properties. Temperature dependence of (a) electrical conductivity, (b)

Seebeck coefficient, (c) power factor, (d) thermal conductivity, (e) lattice thermal conductivity, and (f) average zT values of the $(\text{Bi,Sb})_2\text{Te}_3$ alloys before and after SGF-RM. Some data of previously reported typical $(\text{Bi,Sb})_2\text{Te}_3$ -based materials are also shown in **Fig. 2f**^{7,19,27,40,41}.

Reviewer #3 (Remarks to the Author):

The paper presents a novel process for fabricating the BiSbTe (BST) material, a classical thermoelectric system. It is reported that the BST sintered under a super-gravity condition achieves an exceptionally high zT up to 1.91 that exceeds the value of the hot-pressed nanostructured BST ($zT \sim 1.4$). This claim represents a significant advancement in thermoelectric material development. A careful examination of the fabrication process and characterization details is essential. Here are the inquiries and comments regarding the paper.

1. In a super-gravity and high-temperature firing process, controlling microstructural and compositional uniformity is crucial. This information regarding the mass density, pore distribution, and chemical composition can be obtained by sampling the material at various locations along the gravity-field direction.

Response: Thanks for your kind suggestion. In order to clearly observe the microstructure of samples, some typical polished surface SEM images and the corresponding chemical compositions, measured density at different locations (top, middle, bottom) along the gravity-field direction were shown in the revised Fig. S15d-f. The alloy composition, pore distribution and measured density did not show obvious difference along the gravity-field direction for both BST and BST-1 samples. The word “fractured” in line 1 on page 14 was deleted. The sentence “From Fig. S15d-f, the pores distribution, chemical composition and the measured density did not show obvious difference along the gravity-field direction for both BST and BST-1 samples, showing good microstructural and compositional uniformity along the gravity-field direction.” was added before the sentence “These micropores and grain boundaries also helped decrease the thermal conductivity” in lines 2-3 on page 14.

Fig. S15 The typical SEM images, chemical compositions and measured density at different locations (top, middle, bottom) along the gravity-field direction for BST and BST-1 samples. (a) The fractured surface SEM images of BST (lower magnification), (b) The fractured surface SEM images of BST-1 (lower magnification), (c) The fractured surface SEM images of BST-1 (higher magnification), (d) The polished surface SEM images and EDS results of BST, (e) The polished surface SEM images and EDS results of BST-1, (f) measured density at different locations for BST and BST-1. The chemical compositions and density did not show obvious difference along the gravity-field direction.

2. The reproducibility testing was done on the materials fabricated by using different powder size (BST-S) and firing durations (BST-1 & BST-10). I thought the reproducibility test should be done on multiple samples prepared with the same process. Although the thermoelectric properties of the BST-S, BST-1, and BST-10 look comparable in Fig. S6 due to plot scaling, the variations in each property indeed are significant. A thorough analysis of the experimental variations, including the sampling plan,

should be provided.

Response: We did some more BST-1-R samples with the same fabrication process with BST-1 sample and measured their thermoelectric properties. The results were shown in the updated Fig. S8 in the supporting information. The inset image in Fig. S8d showed the average zT value of all the 9 BST-1 samples, the estimated standard error of zT values is less than 5% (error bars). The thermoelectric properties of the 9 BST-1-R samples showed good reproducibility.

Fig. S8 Reproducibility of the thermoelectric properties for more BST-1-R samples with the same fabrication process with BST-1. Temperature dependence of (a) electrical conductivity, (b) Seebeck coefficient, (c) thermal conductivity, (d) zT values, inset image gives the average value of all the BST-1-R samples, which shows excellent reproducibility. The error bars represent the estimated standard error of zT values (<5%).

3. The compositional variation among the BTS-S, BTS-1, and BTS-10 sample is quite significant, as listed in Table S2. It seems to me the raw powder size plays a role during the firing process. The BST-

S with a small power size received more loss in Te composition.

Response: Raw materials with smaller power size showed larger specific surface area, resulting in more volatilization of Te elements during the preparation process. The experimental quartz tubes containing small particle samples broke during the preparation process due to the higher Te vapor pressure. Therefore, the effect of powder size on the chemical composition and thermoelectric properties has not been studied. It is worth noting that the samples with larger powder size used in the article showed less Te volatilization. Although the compositional variation among all the samples was observed (Fig. S15, Table 1), the thermoelectric properties of different samples showed good consistency (Fig. S7, S8).

4. The BST materials after the SGF-RM process appear to have a much lower mass density (~87% of theoretical density). It definitely contributes to a low thermal conductivity. I am curious why the BST-1 exhibits a much higher electrical conductivity than the BST (98% of theoretical density). A physical explanation is necessary.

Response: Thank you for this suggestion. The higher electrical conductivity is considered to be due to increased carrier concentration. The detailed discussion are as follows:

The sentences “The enhanced microstructural defects increased the phonon-scattering and carrier-scattering, resulting in the decrease of lattice thermal conductivity (Fig. 2e) and carrier mobility (Table 1). Based on the equation⁶⁴, $\sigma = ne\mu$ (σ is the electrical conductivity, n is the carrier concentration, e is the electron charge, μ is the carrier mobility), two competing factors of carrier concentration (n) and mobility (μ) determine the electrical conductivity. The enhanced microstructural defects contributed to the decrease in carrier mobility (μ), while the evaporation of excess Te increased the carrier concentration (n). Because the enhanced carrier concentration contributed more to the electrical conductivity, the electrical conductivity increased instead of decreased after SGF-RM (Fig. 2a).” was added before the sentence “To better understand the main factor responsible for the reduced lattice thermal conductivity, the effective medium theory (EMT) and the Debye–Callaway model were used to analyze the contributions of the absence of thermal conduction within the micropores and the various phonon scattering mechanisms, respectively.” in lines 10-13 on page 14.

5. How is the distribution of micro-pores inside the BST materials after SGF-RM process? Is it possible

to have a gradient distribution along the gravity direction, as shown in Fig. 1B. If so, it may affect the thermoelectric properties of the samples. It is suggested to perform microstructural and PALS measurement at different of the material along the gravity-field direction.

Response: In order to clearly observe the distribution of micro-pores inside the sample after SGF-RM process, we did the microstructural and positron annihilation lifetime spectra (PALS) measurement. Some typical polished surface SEM images at different locations (top, middle, bottom) along the gravity-field direction were supplied. The corresponding chemical compositions and measured density at different locations (top, middle, bottom) along the gravity-field direction were also shown in the revised Fig. S15d-f. The pore distribution, chemical compositions and density did not show obvious difference along the gravity-field direction. PALS are sensitive to the nano-sized micro-pores. The measured positron annihilation spectra (Fig. S12) were decomposed into three lifetimes, τ_1 , τ_2 , and τ_3 , with corresponding intensities I_1 , I_2 , and I_3 , respectively, using the LT9.0 software (**Table 2**). The sentences “Based on the positron annihilation lifetime spectra (PALS) at different locations (top, middle, bottom) along the gravity-field direction for BST-1-R sample (Fig. S12, Table 2), τ_1 and I_1 values did not show obvious difference, showing microstructural defects distributed uniformly along the gravity-field direction for BST-1-R sample. The second-lifetime component τ_2 was much longer than τ_1 due to positron trapping and annihilation at several large vacancy clusters or micropores. The lifetime τ_2 at top section are higher than that at bottom section, while the corresponding intensities I_2 are lower than that at bottom section, which indicated that micro-pores at top section showed larger sizes but lower density. In general, the distribution of the larger pores and microstructural defects are uniform in the matrix, while, the size and density of micro-pores show a little inhomogeneous along the gravity-field direction. It is worth noting that this kind of microstructural feature did not obviously affect the thermoelectric properties. The thermoelectric properties of many samples showed good reproducibility (Fig. S7, S8). In short,” were added before the sentence “Positron annihilation measurements showed that more anti-site defects, dislocations, and micropores were introduced in the $(\text{Bi,Sb})_2\text{Te}_3$ alloys after SGF-RM” in lines 10-11 on page 11.

Fig. S15 The typical SEM images, chemical compositions and measured density at different locations (top, middle, bottom) along the gravity-field direction for BST and BST-1 samples. (a) The fractured surface SEM images of BST (lower magnification), (b) The fractured surface SEM images of BST-1 (lower magnification), (c) The fractured surface SEM images of BST-1 (higher magnification), (d) The polished surface SEM images and EDS results of BST, (e) The polished surface SEM images and EDS results of BST-1, (f) measured density at different locations for BST

and BST-1. The chemical compositions and density did not show obvious difference along the gravity-field direction.

Fig. S12 The PALS spectra. (a) The PALS spectra of BST and BST-1 samples, (b) The PALS spectra at different locations (top, middle, bottom) along the gravity-field direction of BST-1-R sample.

Table 2 Positron annihilation lifetime spectroscopy (PALS) data of BST, BST-1 and BST-1-R.

Specimen	τ_1 (ns)	I_1 (%)	τ_2 (ns)	I_2 (%)	τ_3 (ns)	I_3 (%)
BST	0.1521	26.3	0.3209	71.6	1.286	2.06
BST-1	0.1685	31.1	0.3344	67.7	1.534	1.25
BST-1-R top	0.2076	60.90	0.3922	38.4	1.870	0.70
BST-1-R middle	0.2076	60.91	0.3870	38.4	1.762	0.69
BST-1-R bottom	0.2056	57.98	0.3758	41.3	1.765	0.73

6. The dislocation density of BST-1 was estimated to be $7 \times 10^{12} \text{ cm}^{-2}$. Please provide the dislocation density of the original BST sample for comparison.

Response: In order to observe the dislocation distribution, the strain field images of the BST sample before SGF were shown in Fig. 3j-I and the inverse fast Fourier transform (IFFT) images were shown in Fig. S13d-f. The dislocation density in the BST sample was roughly estimated to be about $10^5 - 10^6 \text{ cm}^{-2}$. Mion et al. reported that dislocation densities above 10^7 cm^{-2} have a significant effect on the lattice thermal conductivity of GaN¹. Thus, the effect of the dislocation density on the lattice thermal conductivity of BST was not considered when calculating the thermal transport properties based on

the Debye-Callaway' s model (Fig. 1e).

Fig. S13 TEM images of the BST sample. (a, b) Low-magnification in different regions, (c) high-magnification, (d–f) Inverse fast Fourier transform (IFFT) images in the (110), (2 $\bar{1}0$) and ($\bar{1}20$) planes obtained from the area marked by the white rectangle in Fig. S12c.

In summary, the results presented in this paper are impressive. However, the zT values were determined from individual measurements of transport properties, which may introduce some variations. Therefore, a detailed analysis of the experimental measurements is necessary to substantiate any claims of a record-high zT value.

Response: Thanks for your kind suggestion. We added more detailed characterization and analysis in the revised version. In order to clearly observe the microstructure of samples, positron annihilation lifetime spectra (PALS), surface SEM images and the corresponding chemical compositions, density at different locations (top, middle, bottom) along the gravity-field direction were measured, which showed good microstructural and compositional uniformity along the gravity-field direction. The enhanced microstructural defects after SGF-RM increased the phonon-scattering and carrier-scattering, resulting in the decrease of lattice thermal conductivity and carrier mobility. However, the increased the carrier concentration due to excess Te evaporation contributed more to the electrical conductivity.

As a result, the zT value greatly enhanced.

References

1. Mion, C., Muth, J. F., Preble, E. A. & Hanser, D. Thermal conductivity, dislocation density and GaN device design. *Superlattice microst.* **40**, 338-342 (2006).

Dear Reviewers,

Many thanks for your report on our manuscript entitled: “Ultrahigh thermoelectricity obtained in classical BiSbTe alloy processed under super-gravity” (Manuscript ID: NCOMMS-25-09159A). We really appreciate the professional comments. Our point-to-point responses for all reviewers are listed as below:

Reviewer #2 (Remarks to the Author):

The authors have revised the manuscript considerably and can be accepted in the current form.

Response: Thanks a lot for your kind approval.

Reviewer #3 (Remarks to the Author):

I think the revision has partially addressed some of my considerations on the original manuscript regarding the uniformity of material properties. However, some points still require further clarification.

1. The authors prepared multiple BST-1-R samples to test the reproducibility of the process proposed. It is claimed that the standard error of zT values is less than 5%. Since the zT is calculated from the individual transport property measurements, it is difficult to be convinced that the accumulated errors in zT are less than 5%. A rigorous error estimation based on the respective properties should be provided.

Response: Thanks for your kind suggestion. As shown in the main text, the dimensionless figure of merit $zT = \alpha^2 \sigma T / \kappa$, where α , σ , κ and T are the Seebeck coefficient, electrical conductivity, thermal conductivity and absolute temperature, respectively. The values of α and σ can be directly measured. The thermal conductivity $\kappa = \lambda d C_p$ (λ is thermal diffusivity coefficient, d is density, C_p is heat capacity). So, the uncertainty of zT can be expressed as¹:

$$\frac{\Delta(zT)}{(zT)} = 2\frac{\Delta\alpha}{\alpha} + \frac{\Delta\sigma}{\sigma} + \frac{\Delta\lambda}{\lambda} + \frac{\Delta C_p}{C_p} + \frac{\Delta d}{d} + \frac{\Delta T}{T} \quad (1)$$

Usually, the uncertainties of commercial instruments are $\pm 4\%$ for α , $\pm 3\%$ for σ , $\pm 3\%$ for λ , $\pm 5\%$ for C_p ². The measured error of Archimedes method and thermocouple temperature measurement are very small. So, $\frac{\Delta d}{d} + \frac{\Delta T}{T}$ in equation (1) can be neglected. The estimated accumulated measured error (uncertainty) of zT is about $\pm 15\%$.

However, the repeatability, which is the closeness of agreement between successive measurements carried out under the same condition, can be much better than the accuracy when using those commercial instruments. Thus, repeatability is more important and useful when comparing TE properties in experiments. In Fig. S8 in the supporting information, we calculated and shown the standard error of zT values (<5%) for all the BST-1-R samples, which shows excellent reproducibility.

The formula for the standard error of the sample mean is³:

$$SE = \frac{\sqrt{\frac{1}{n-1} \sum_{i=1}^n (x_i - \bar{x})^2}}{\sqrt{n}} \quad (2)$$

Where, n is the number of samples, the numerator of formula (2) refers to the standard deviation. The standard error is the standard deviation of the distribution of a sample statistic, reflecting the precision of the sample estimate. A smaller standard error indicates a more precise estimate.

We updated the “methods” section in the main text. The sentence “Usually, the uncertainties of commercial instruments are $\pm 4\%$ for α , $\pm 3\%$ for σ , $\pm 3\%$ for λ , $\pm 5\%$ for C_p ⁷⁴. The measured error of Archimedes method and thermocouple temperature measurement are very small. So, the measured error from d and T can be neglected. Combining the electrical conductivity, Seebeck coefficient, and thermal conductivity obtained from the measurements, the estimated accumulated measured error (uncertainty) of zT is about $\pm 15\%$.” was added after the sentence “The transport properties in the parallel direction were repeated (Fig. S7, S8).” in lines 18-19 on page 30.

2. In the rebuttal letter, the authors noted that the quartz tubes containing the small particle samples (BST-S) broke during the preparation process due to the higher Te vapor pressure. The vigorous evaporation of Te may explain the significantly lower Te content in the BST-S compared to the BST-1 and BST-10 samples. As proposed by the authors, excess Te evaporation facilitates the formation of antisite defects and increases the carrier concentration of the BST-based samples. It is thus expected that the BST-S sample should exhibit a higher carrier concentration and a lower Seebeck coefficient.

This expectation seems to align with the results shown in Fig. 2, but is contradictory to the Hall measurement results presented in Table 1.

Response: As discussed in the rebuttal letter, raw materials with smaller particle size showed larger specific surface area, resulting in more volatilization of Te elements during the preparation process (Fig. 1d, BST-S sample with particle size of 0.6-1.43 mm). The experimental quartz tubes containing much more smaller particle samples (Fig. SS1e. Sample 1 with particle size of 0.25-0.15 mm, Fig. SS1f. Sample 2 with particle size of 0.075-0.09 mm) broke during the preparation process due to the much more higher Te vapor pressure. We have to say that we made a very stupid and simple mistake during the process of revising the manuscript. The ICP-OES results of BST-S and BST-1 samples in Table 1 on page 10 were wrong. The total content (mol %) of Bi, Sb and Te for BST-S and BST-1 samples was not 100% (Table SS1). In order to check the ICP-OES results, we re-measured two other samples and showed the results in Table SS2. The re-measured ICP-OES results for BST-S and BST-1 are $\text{Bi}_{9.99}\text{Sb}_{29.27}\text{Te}_{60.74}$ and $\text{Bi}_{10.20}\text{Sb}_{28.45}\text{Te}_{61.35}$, respectively. Accordingly, the data in Table 1 were undated in the revised version. Clearly, the content of tellurium in these samples (BST-1, BST-10 and BST-S) does not vary much. As discussed in the raw manuscript, the carrier concentration increased and the mobility decreased due to the evaporation of excess telluride after SGF-RM. As a result, the electrical conductivity and power factor showed increasing trend, which was confirmed by all three samples (BST-1, BST-10 and BST-S) in Fig. 2. It is worth noting that the Hall measurement results of BST-S showed deviation in Table 1. In fact, the Seebeck coefficient (α) and electrical conductivity (σ) are measured by using the Seebeck Coefficient/Electrical Resistance Measuring System (ZEM-3, Ulvac-Riko) and the Hall coefficient (R_H) is measured by a Hall measurement system (ResiTest 8340DC, Toyo, Japan) via the Van der Pauw method. The values of electrical conductivity in Fig. 2 and Table 1 are obtained from different testing equipment and different measurement method, which maybe contribute to the inconsistency of the electrical properties for BST-S sample. In some literatures^{4,5}, different electrical properties were also reported by using different measurement methods.

Fig. SS1. The real photos of the different samples.

Table SS1 The raw ICP-OES results of all the samples.

samples	ICP-OES results	Bi (mol %)	Sb (mol %)	Te (mol %)	Total (mol %)	Remark
BST	Bi _{8.51} Sb _{28.92} Te _{62.57}	8.51	28.92	62.57	100.00	Correct
BST-S	Bi _{9.69} Sb _{30.31} Te _{56.99}	9.69	30.31	56.99	96.99	Incorrect
BST-1	Bi _{10.56} Sb _{29.44} Te _{62.17}	10.56	29.44	62.17	102.17	Incorrect
BST-10	Bi _{8.57} Sb _{29.12} Te _{62.30}	8.57	29.12	62.30	99.99	Correct

Table SS2 The re-measured ICP-OES results of BST-S and BST-1 samples.

Test element	Bi, Sb, Te
Data calculation formula	$Cx(ug/kg) = \frac{C_0(ug/L) * f * V_0(mL) * 10^{-3}}{m(g) * 10^{-3}} = \frac{C_1(ug/L) * V_0(mL) * 10^{-3}}{m(g) * 10^{-3}}$ $W(\%) = \frac{Cx(ug/kg)}{10^9} * 100\%$

Sample	Constant volume V_0 (mL)	Sample weight m_0 (g)	Element	Element concentration in test solution C_0 (ug/L)	Dilution times f	Element concentration in original digestion solution C_1 (ug/L)	Element content of the sample C_x (ug/kg)	Element content of the sample W (wt%)	Element content of the sample M (mol %)
BST-S	0.0451	25	Bi	283.374	1000	283373.5663	157080690.8	15.71%	9.99%
			Sb	48.392	10000	483917.4076	268246900.0	26.82%	29.27%
			Te	105.203	10000	1052026.413	583163200.0	58.32%	60.74%
BST-1	0.0451	25	Bi	101.011	1000	101010.8053	158822020.9	15.88%	10.24%
			Sb	164.190	10000	1641899.431	910143808.8	91.01%	28.45%
			Te	370.913	1000	370913.3507	583197092.3	58.32%	61.35%

Table 1 The ICP-OES results, the corresponding Hall measurement of all the samples before and after SGF-RM.

samples	ICP-OES results	Carrier concentration (10^{19} cm^{-3})	Mobility ($\text{cm}^2/\text{V S}$)	Hall coefficient (cm^3/C)	Electrical resistivity ($10^{-3} \Omega \text{ cm}$)
BST	$\text{Bi}_{8.51}\text{Sb}_{28.92}\text{Te}_{62.57}$	1.37	388.98	0.46	1.17
BST-S	$\text{Bi}_{9.99}\text{Sb}_{29.27}\text{Te}_{60.74}$	2.08	279.01	0.30	1.03
BST-1	$\text{Bi}_{10.20}\text{Sb}_{28.45}\text{Te}_{61.35}$	2.42	358.62	0.26	0.72
BST-10	$\text{Bi}_{8.57}\text{Sb}_{29.12}\text{Te}_{62.30}$	2.69	315.06	0.23	0.74

3. It is curious to me why the BST-S, with such a high Te content deviation, still

possesses comparable transport properties compared to the BST-1 and BST-10 samples.

Response: As discussed in the above question 2, we made a mistake in the ICP-OES measurement of BST-S and BST-1 samples. The re-measured ICP-OES results of BST-S and BST-1 are $\text{Bi}_{9.99}\text{Sb}_{29.27}\text{Te}_{60.74}$ and $\text{Bi}_{10.20}\text{Sb}_{28.45}\text{Te}_{61.35}$, respectively. The chemical composition of the BST-1, BST-10 and BST-S samples did not show so obvious difference. This is mainly because all samples are sealed in quartz tubes and the melting time was also short during the preparation process. As discussed in the raw manuscript, the carrier concentration increased due to the evaporation of excess telluride after SGF-RM. As a result, the electrical conductivity increased and the Seebeck coefficient decreased. Accordingly, the power factor showed increased trend after SGF-RM. It is worth mentioning that the power factor of these samples (BST-1, BST-10, BST-S) do not differ much because the electrical conductivity increases while the Seebeck coefficient decreases after SGF-RM.

References

1. Li, Z.-Y. & Li, J.-F. Fine-Grained and Nanostructured $\text{AgPb}_m\text{SbTe}_{m+2}$ Alloys with High Thermoelectric Figure of Merit at Medium Temperature. *Adv. Energy Mater.* **4**, 201300937 (2014).
2. Wei, T.-R. *et al.* How to Measure Thermoelectric Properties Reliably. *Joule* **2**, 2183-2188 (2018).
3. Hess, A. S. & Hess, J. R. Understanding standard deviations and standard errors. *Transfusion* **56**, 1259-1261 (2016).
4. Zhao, W. *et al.* Superparamagnetic enhancement of thermoelectric performance. *Nature* **549**, 247-251 (2017).
5. Pei, Y. *et al.* Convergence of electronic bands for high performance bulk thermoelectrics. *Nature* **473**, 66-69 (2011).

Dear Reviewers,

Many thanks for your report on our manuscript entitled: “Ultrahigh thermoelectricity obtained in classical BiSbTe alloy processed under super-gravity” (Manuscript ID: NCOMMS-25-09159B). We really appreciate the professional comments. Our point-to-point responses for all reviewers are listed as below:

Reviewer #3 (Remarks to the Author):

I believe my comments and suggestions have been adequately addressed in the revised manuscript and the rebuttal letter. I am pleased to recommend the publication of this paper.

Response: Thanks a lot for your kind approval.